



**Short-term fate of intertidal microphytobenthos carbon under enhanced nutrient**
**availability: A [13]C pulse-chase experiment**
Philip M. Riekenberg[1,a*], Joanne M. Oakes[1], Bradley D. Eyre[1]
[1]Centre for Coastal Biogeochemistry, Southern Cross University, PO Box 157, Lismore,
NSW, 2480, Australia
[a]Present address: NIOZ Royal Netherlands Institute for Sea Research, Department of Marine
Microbiology and Biogeochemistry, PO Box 59, 1790AB Den Burg
[*] Corresponding author: phrieken@gmail.com, (0031) 222 369 409
Keywords: benthic microalgae, bacteria, biomarker, nutrients
Running head: Intertidal carbon processing





ABSTRACT:
Shallow coastal waters in many regions are subject to nutrient over-enrichment.
Microphytobenthos (MPB) can account for much of the carbon (C) fixation in these
environments, depending on the depth of the water column, but the effect of enhanced nutrient
availability on the processing and fate of MPB-derived C is relatively unknown. In this study,
MPB were labeled (stable isotope enrichment) in situ using $^{13}$C-sodium bicarbonate. The
processing and fate of the newly-fixed MPB-C was then traced using ex situ incubations over
3.5 d under different concentrations of nutrients ($NH_4^+$ and $PO_4^{3-}$: ambient, 2× ambient, 5×
ambient, and 10× ambient). After 3.5 d, sediments incubated with increased nutrient
concentrations (amended treatments) had increased loss of $^{13}$C from sediment organic matter
as a portion of initial uptake (95% remaining in ambient vs 79-93% for amended treatments)
and less $^{13}$C in MPB (52% ambient, 26-49% amended), most likely reflecting increased
turnover of MPB-derived C supporting increased production of extracellular enzymes and
storage products. Loss of MPB-derived C to the water column via dissolved organic C was
minimal regardless of treatment (0.4-0.6%). Loss due to respiration was more substantial, with
effluxes of dissolved inorganic C increasing with additional nutrient availability (4% ambient,
6.6-19.8% amended). These shifts resulted in a decreased turnover time for algal C (419 d
ambient, 134-199 d amended). This suggests that nutrient enrichment of estuaries may
ultimately lead to decreased retention of carbon within MPB-dominated sediments.



## 1.0 Introduction

Intertidal sediments are important sites for the processing of carbon (C) within

estuaries, producing, remineralizing, and transforming considerable amounts of organic

material prior to its export to the coastal shelf (Bauer et al. 2013). Algal production is a key

source of C within the coastal zone, and is primarily derived from microphytobenthos (MPB)

in shallow photic sediments (Hardison et al. 2013; Middelburg et al. 2000). In addition to

algal cells being a labile carbon source, MPB exude large amounts of carbohydrates as

extracellular polymeric substances (EPS) (Goto et al. 1999) that allow for vertical migration

and enhance sediment stability (Stal 2010). A better understanding of the carbon pathways

utilized during processing of algal cells and exudates within sediments is important for

determining the quality and quantity of carbon exported from estuarine waters to continental

shelves.

Application of stable isotope tracers cause isotopic enrichments that usually serve to

make negligible the variability associated with fractionation effects in natural abundance

stable isotope techniques. As such, stable isotope tracers have been useful for elucidating

pathways for the processing and loss of MPB-derived C within estuarine sediments. Loss

pathways for MPB-derived C include resuspension (Oakes and Eyre 2014), fluxes of

dissolved inorganic C (DIC) due to mineralization and respiration (Evrard et al. 2012; Oakes

et al. 2012), fluxes of dissolved organic C (DOC) comprised of microbial exudates and

products from cell lysis (Oakes et al. 2010a), and direct production of $CO_2$ (Oakes and Eyre

2014). Stable isotope tracer studies have also enabled quantification of the trophic transfer

(Middelburg et al. 2000; Miyatake et al. 2014; Nordström et al. 2014; Oakes et al. 2010a) and





flux of newly produced C from sediments (Andersson et al. 2008; Oakes et al. 2012; Van
Nugteren et al. 2009).

When $^{13}$C is combined with analysis of phospholipid-linked fatty acids (PLFAs), it

becomes possible to trace C transfer into individual microbial groups that account for the
living biomass within sediment organic C (OC) (Drenovsky et al. 2004; Hardison et al. 2011;
Oakes et al. 2012; Oakes and Eyre 2014; Spivak 2015). This allows for the quantification of
microbial transfers of newly produced algal C between MPB and bacteria and the relative
contributions of MPB and bacteria to microbial biomass in sediment OC. This technique has
shown that EPS produced by MPB is readily utilized as a C source for heterotrophic bacteria
(Oakes et al. 2010; van Oevelen et al. 2006). Pathways for processing of MPB-derived C have
been reasonably well described, but the response of these pathways to local environmental
changes remains a significant knowledge gap.

A major source of environmental change in coastal systems is nutrient over-

enrichment (Cloern et al. 2001), which may affect the assimilation and flux pathways of
MPB-derived carbon through 1) increased microbial biomass resulting from relaxation of
nutrient limitation, 2) increased algal production that drives elevated heterotrophic processes
as bacteria utilize newly produced C, and 3) increased loss of C as DIC via respiratory
pathways as heterotrophic processes dominate. MPB are able to use both porewater and water
column nutrients, and although MPB biomass can increase with elevated nutrient availability
(Armitage and Fong 2004; Cook et al. 2007), this is not always the case, with multiple studies
finding no corresponding increase in MPB biomass (Alsterberg et al. 2012; Piehler et al. 2010;
Spivak and Ossolinski 2016). Processing and mineralization of C are significantly affected by
changes in the relationship between MPB and bacteria (Evrard et al. 2012). Both EPS



production and bacterial utilization of newly produced EPS may decrease with increasing
nutrient availability (Cook et al. 2007). Increased autochthonous production driven by nutrient
enrichment can lead to increased heterotrophy, as newly produced organic matter is
mineralized (Fry et al. 2015), resulting in increased DIC production. Increased
remineralization of newly produced MPB-C will result in greater loss of DIC from intertidal
sediment via bacterial respiration (Hardison et al. 2011).
In this $^{13}$C pulse-chase study we aimed to quantify the processing pathways of MPB-
derived C within subtropical intertidal sediments and to determine how this is affected by
increased nutrient loading. The in situ MPB community was used to provide a pulse of labeled
MPB-C of similar quantity and quality to normal production. Even application of separate
label applications for each plot prior to incubation served to isolate the subsequent effect of
increased nutrient availability on the processing of MPB-derived C. Pathways considered
included transfer through sediment compartments (MPB, bacteria, uncharacterized and
sediment OC), and loss via fluxes of DOC and DIC. We expected increased nutrient
availability to stimulate MPB production of EPS after initial labeling, resulting in decreased
turnover times for MPB-C as well as a shift towards dominance of heterotrophic processes as
bacteria utilize this additional labile C. We further hypothesized that enhanced heterotrophy
would increase loss of newly fixed algal-derived C via respiration as DI$^{13}$C. Incorporation of
$^{13}$C into biomarkers should reflect the shift towards heterotrophy, with quicker shifts towards
increased bacterial utilization of newly produced algal C corresponding with increased
nutrient load. Both DIC and DOC should be significant loss pathways for newly produced
algal C as labile OM is readily processed by heterotrophs.
**2.0 Methods**



**2.1 Study site**


The study site was a subtropical intertidal shoal ~ 2 km upstream of the mouth of the
Richmond River estuary in New South Wales, Australia (28°52'30"S, 153°33'26"E). The
6900 km² catchment has an annual rainfall of 1300 mm (McKee et al. 2000) and an average
flow rate of 2200 ML d⁻¹ (daily gauged flow adjusted for catchment area, averaged over years
for which data was available; 1970–2013). Although the Richmond River Estuary has highly
variable flushing, salinity, and nutrient concentrations associated with frequent episodic
rainfall events and flooding (Eyre 1997; Mckee et al. 2000), this study was undertaken during
a dry period. The site experiences semidiurnal tides with a range of ~2 m. Samples were
collected in summer January 2015 with average site water temperature of $25.6 \pm 2.3°C$.
Sediment at depths of 0-2 cm, 2-5 cm and 5-10 cm was dominated by fine sand (66%-73%),
and sediment across 0-10 cm had an organic C content of $17.5 \pm 0.02$ mol C m⁻². Sediment
molar C:N was lowest at 2-5 cm, but comparable across all other depths (top scrape (TS) 14.4
$\pm$ 1.6, 0-2 cm 17.2 $\pm$ 1.7, 2-5 cm 10.9 $\pm$ 0.5, 5-10 cm 16.2 $\pm$ 2.2).

**2.2 Overview**


We labeled MPB with ¹³C via in situ application of ¹³C-labeled sodium bicarbonate to
exposed intertidal sediments. Unincorporated ¹³C was flushed from the sediment during the
next tidal inundation of the site. Sediment cores were collected and incubated in the laboratory
over 3.5 d under four nutrient enrichment scenarios (ambient, minimal, moderate, and
elevated) using pulsed nutrient additions. Incubation of cores ex situ allowed for explicit
control of nutrient additions and examination of the short-term processing and fate (loss to
overlying water) of MPB carbon. Sediments remained inundated during incubations with
minimal water exchange, as might be expected during neap tide at this site. Inundation also



served to minimize C loss via physical resuspension and export while we were examining
sediment processing.

## 2.3 $^{13}$C-labeling

Bare sediment within a 2 m$^2$ experimental plot was $^{13}$C-labeled when sediments were

first exposed during the ebbing tide in the middle of the day by using motorized sprayers to
evenly apply 99% NaH$^{13}$CO$_3$ onto individual 400 cm$^2$ subplots, closely following the method
outlined in Oakes and Eyre (2014). Label applications were prepared using NaCl-amended
Milli-Q matching site salinity (34.6), and 20 ml aliquots (1.7 mmol $^{13}$C) were applied to each
individual subplot, resulting in application of 42.5 mmol $^{13}$C m$^{-2}$. The use of individual
aliquots of label ensured even $^{13}$C application across the sediment surface. Assimilation of
label by the sediment community occurred over ~4 hours with average light exposure of 1376
µE m$^{-2}$ s$^{-1}$, before tidal inundation removed the majority of unincorporated $^{13}$C. Removal was
confirmed by loss of 99.0% of the applied $^{13}$C from treatment applications within initial cores
sampled in the field.

## 2.4 Collection of sediment cores

Prior to label application, 3 cores (9 cm diameter, 20 cm depth) were collected from

unlabeled sediment surrounding the treatment plots and immediately extruded and sectioned
(0-0.2 cm (top scrape, TS), 0.2-2 cm, 2-5 cm, and 5-10 cm) to provide control natural
abundance sediment OC δ$^{13}$C values for sediment depths within the study site. Eleven hours
after label application, at low tide, 35 sediment cores were similarly collected from the labeled
plot using Plexiglas core liners. Immediately, three cores were extruded and sectioned as
above to determine initial $^{13}$C uptake and grain size distribution for all sediment depths, and
chlorophyll-α (Chl-α) concentrations in 0-1 cm sediments. All samples were placed within





plastic zip-lock bags, transported to the laboratory on ice, and stored frozen in the dark (-20°
C). Plexiglas plates were used to seal the bottom of the core liners, and cores for incubation
were then transported to the laboratory within 2 hours of sampling. Site water (400 L) was
collected and transported to the laboratory for use in incubations.
**2.5 Nutrient amendment**

Pulsed applications of nutrients for each treatment amendment were used to mimic a

range of nutrient concentrations without exceeding sediment capacity for uptake. The
treatment tanks were set up at ambient concentration (site water, DIN of $2.5 \pm 0.04$ µmol N L$^-$
$^1$, measured on incoming tide), and with N (NH$_4^+$) and P (H$_3$PO$_4$) amendment to site water at
2× (minimal treatment), 5× (moderate treatment) and 10× (elevated treatment) average water
column concentrations near the study site (4 µmol L$^{-1}$ NH$_4^+$ and 5 µmol L$^{-1}$ PO$_4^{3-}$; Eyre (1997;
2000). To allow thorough mixing, the initial pulse of nutrients was added to both incubation
tanks and bags holding replacement water for sampling, one hour prior to cores being
transferred into the incubation tanks. An additional pulse of NH$_4^+$ was applied to incubation
tanks at 1.5 d (after sample collection) to mimic the nutrient availability that occurs with
regular inundation of tidal sediments. Silica (Si) was also added to all incubation tanks at 2.5 d
(after sample collection) to ensure that isolation of the benthic diatom-dominated sediment
from regular water turnover did not result in secondary limitation of Si. There was no
significant accumulation of NH$_4^+$ within treatment tank water, as nutrients were readily
processed (Supplemental Fig. 1).
**2.6 Benthic flux incubations**

In the laboratory, cores were fitted with magnetic stir bars positioned 10 cm above the

sediment surface, filled with ~2 L of site water, and randomly allocated to one of the four 85





L treatment tanks (Ambient, Minimal, Moderate, Elevated; eight cores per treatment). Water
in the treatment tanks and cores was continuously recirculated, held at in situ temperature (25
$\pm$ 1$^{o}$C) by a chiller on each tank, and aerated. Cores were stirred via a rotating magnet at the
center of each treatment tank, which interacted with the magnetic stir bars. Stirring occurred at
a rate below the threshold for sediment resuspension (Ferguson et al. 2003). Three sodium
halide lamps suspended above the treatment tanks provided 824 $\pm$ 40 $\mu$E m$^{-2}$ s$^{-1}$ to the
sediment/water interface within the cores on a 12 h light/12 h dark cycle which approximated
the average light level measured at the sediment surface during inundation (941.4 $\pm$ 139 $\mu$E m$^{-}$
$^{2}$ s$^{-1}$). Cores were allowed to acclimate in tanks for 6 h prior to the start of incubation. Cores
remained open to the tank water until 30 min before sampling, when clear Plexiglas lids were
fitted to each core liner to seal in overlying water within the core for the duration of the
incubation (~16 h). Dissolved oxygen ($\pm$ 0.01 mg L$^{-1}$) and pH ($\pm$ 0.002 pH) were measured
optically and electrically (Hach HQ40d multi-parameter meter) via a sampling port in the lid.
Initial samples were taken 30 min after closure of the lids, dark samples were taken after ~12
hours incubation with no light, and light samples were taken after 3 hours of illumination
following the end of the dark period. During sampling, 50 ml of water was syringe-filtered
(precombusted GF/F) into precombusted 40 ml glass vials with Teflon coated septa, killed
with HgCl$_2$ (20 $\mu$L saturated solution), and refrigerated prior to analysis for concentration and
$\delta^{13}$C of DIC and DOC. Sample water was simultaneously replaced by water held in
replacement bags as sampling occurred at each time point. No samples were collected for
analysis of gaseous CO$_2$ fluxes from exposed sediments, as this was previously determined to
be a negligible pathway for loss of MPB-C at this site (Oakes and Eyre 2014). At the end of
sampling for the light period, cores were extruded, sectioned and sampled for Chl-$\alpha$ in the



same manner as control cores and stored frozen ($-20^{\circ}$C). Eight cores (two cores per treatment)
were sampled in this manner for water column fluxes, PLFAs, and sediment OC after 1.5 d,
2.5 d and 3.5 d of incubation. Additionally, 8 cores (two cores per treatment) were sampled
for only PLFAs and sediment OC at 0.5 d of incubation.
**2.7 Sample analysis**

Chl-$\alpha$ was measured by colorimetry (Lorenzen 1967) for each core (0-1 cm depth).

MPB-C biomass was calculated assuming a C:Chl-$\alpha$ ratio of 40, within the range reported for
algae in Australian subtropical estuaries (30-60 Ferguson et al. 2004; Oakes et al. 2012).
Biomass measurements utilizing Chl-α were used to compare biomass across controls and
treatments and were not utilized in calculations for uptake of $^{13}$C into MPB or bacteria using
PLFAs. Bacterial C biomass for controls was estimated based on MPB-C biomass derived
from Chl-α and the ratio of MPB to bacterial biomass obtained from PLFA analysis of the
control cores (n=3).

Sediment samples were lyophilized, loaded into silver capsules, acidified (10% HCl),

dried ($60^{\circ}$C to constant weight), and analyzed for %C and $\delta^{13}$C using a Thermo Flash
Elemental Analyzer coupled to a Delta V IRMS via a Thermo ConFlo IV. Samples were run
alongside glucose standards that are calibrated against international standards (NBS 19 and
IAEA ch6). Precision for $\delta^{13}$C was 0.1‰ with decreasing precision for enrichments above
100‰.

PLFAs specific to bacteria (i + a 15:0) were used as biomarkers for this group.

However, although visual analysis confirmed the presence of a large number of pennate
diatoms at the study site and diatom-specific PLFAs (e.g. 20:5(n-3)) were detected,





chromatographic peaks for these PLFAs were sometimes indistinct. The 16:1(n-7) PLFA,
which represents 27.4% of total diatom PLFAs (Volkman et al. 1989), was therefore used as a
biomarker for diatoms, following correction for contributions from gram-negative bacteria and
cyanobacteria as described below and in Oakes et al. (2016), as it was consistently present
across all samples. Extraction of PLFAs used 40 g of freeze-dried sediment and a modified
Bligh and Dyer technique. Sediment was spiked with an internal standard (500 µL of 1 mg ml$^{-1}$
tridecanoic acid, $C_{13}$), immersed in 30-40 ml of a 3:6:1 mixture of dichloromethane (DCM),
methanol, and Milli-Q water, sonicated (15 min), and centrifuged (15 min, 9 $g$). The
supernatant was removed into a separating funnel and the pellet was re-suspended in 30-40 ml
of the DCM:MeOH:Milli-Q mixture, sonicated, and centrifuged twice more to ensure
complete removal of biomarkers. DCM (30 ml) and water (30 ml) were added to the
supernatant, gently mixed, and phases were allowed to separate prior to removal of the bottom
layer into a round bottom flask. The top layer was then rinsed with 15 ml of DCM, gently
shaken, and phases allowed to separate prior to addition to the round bottom flask. This
extract was then concentrated under vacuum and separated using silica solid phase extraction
columns (Grace; 500 mg, 6.0 ml) by elution with 5 ml each of chloroform, acetone, and
methanol. The fraction containing methanol was retained, reduced to dryness under $N_2$,
methylated (3 ml 10:1:1 MeOH:HCl:CHCl$_3$, 80˚C, 2 h), quenched using first 3 ml and then 2
ml of 4:1 hexane:DCM, evaporated to ~ 200 µl under $N_2$, transferred to a GC vial for analysis,
and stored frozen (-20˚C). PLFA concentrations and $\delta^{13}$C values were measured using a non-
polar 60 m HP5-MS column in a Trace GC coupled to a Delta V IRMS with a Thermo Conflo
III interface following the protocol outlined in Oakes et al. (2010a).



DIC and DOC concentrations and $\delta^{13}C$ values were measured via continuous-flow wet
oxidation isotope-ratio mass spectrometry using an Aurora 1030W total organic C analyzer
coupled to a Thermo Delta V isotope ratio mass spectrometer (IRMS) (Oakes et al. 2010b).
Sodium bicarbonate (DIC) and glucose (DOC) of known isotopic composition dissolved in
He-purged Milli-Q were used to correct for drift and verify both concentration and $\delta^{13}C$ of
samples. Reproducibility was $\pm 0.2$ mg L$^{-1}$ and $\pm 0.1$ ‰ for DIC and $\pm 0.2$ mg L$^{-1}$ and $\pm 0.4$ ‰
for DOC.
**2.8 Calculations**
Incorporation of $^{13}C$ into sediment OC, bacteria, and MPB (mmol $^{13}C$ m$^{-2}$) was
calculated as the product of excess $^{13}C$ (fraction $^{13}C$ in sample – fraction $^{13}C$ in control) and
the mass of OC within each pool. For sediment, OC was the product of %C and dry mass per
unit area.
Excess $^{13}C$ for PLFAs was determined only for 0-2 cm, 2-5 cm, and 5-10 cm depths, as
there was inadequate sample mass for the 0-0.2 cm top scrape. Due to limitations of time and
cost, PLFA samples were taken from only one of the two cores incubated for each treatment at
each sampling period. PLFA excess $^{13}C$ for both bacteria and diatoms was the product of
excess $^{13}C$ contained in the PLFA (fraction $^{13}C$ in PLFA in sample – fraction $^{13}C$ in PLFA in
control) and the concentrations of C within respective biomarkers. Concentrations of PLFA C
were calculated from their peak areas relative to the internal $C_{13}$ standard spike. Biomass of
diatoms and bacteria were calculated using the method described by Oakes et al. (2016).
Briefly, bacterial biomass was calculated as:
1.  Biomass$_{bacteria}$ = Biomass$_{i+a15:0}$ / $(a \times b)$



where $a$ represents the average concentration of PLFA (0.056 g C PLFA per g C
biomass; Brinch-Iversen and King 1990) in bacteria and $b$ represents the average fraction of
PLFA accounted for by i+a15:0 within bacteria-dominated marine sediments (0.16, Osaka
Bay, Japan; Rajendran et al. 1994; Rajendran et al. 1993). Biomass estimates for bacteria
calculated using the minimum and maximum fraction values (16-19% for i+a15:0; Rajendron
et al. 1993) resulted in a 16% difference.
For diatoms, a mixing model was used to correct the concentration and $\delta^{13}$C value of
16:1(n-7) for the any contribution from non-diatom sources. Due to the scarcity of
cyanobacteria observed using light microscopy (1000×), low sediment D-/L- alanine ratios
measured previously at this site (as low as 0.0062, Riekenberg et al. 2017), and lack of the
characteristic 18:2(n-6) peak (Bellinger et al. 2009) cyanobacteria were assumed to make a
negligible contribution to the 16:1(n-7) peak. A two-source mixing model was applied to
correct the concentration and $\delta^{13}$C value of the 16:1(n-7) peak for the contribution of gram-
negative bacteria, based on a typical ratio of 18:1(n-7) to 16:1(n-7) for gram-negative bacteria
of 0.7 (Edlund et al. 1985) as previously applied in Oakes et al. (2016). Biomass for diatoms
was calculated using the formula:
2. $\text{Biomass}_{\text{Diatom}} = \text{Biomass}_{\text{corrected16:1(n-7)}} / (c \times d)$
where $c$ is the average fraction of diatom PLFAs accounted for by corrected 16:1(n-7)
(0.67; Volkman et al. 1989) and $d$ is the average PLFA concentration in diatoms (0.035 g
PLFA C per g of C biomass; Middelburg et al. 2000). Biomass estimates for diatoms
calculated using maximum and minimum fraction values for 16:1(n-7) (18-33%; Volkman et
al. 1989) were within 50% of estimates based on the average value.



Fluxes across the sediment-water interface were calculated as a function of incubation

time, core water volume and sediment surface area. Dark flux rates were calculated using
concentration data from the dark incubation period and light flux rates from the light
incubation period. The following parameters were calculated from dark and light rates:

3.  Respiration (R) = Dark DO flux $h^{-1}$

4.  Net primary production (NPP) = Light DO flux $h^{-1}$

5.  Gross primary production (GPP) = NPP + R

6.  Production/respiration (P/R) = GPP x daylight hours (12 h) / R x 24 h (Eyre et al.

2011)

Total $^{13}$C in  DIC and DOC (µmol $^{13}$C) in the overlying water in the sediment core was
calculated for initial, the end of the dark period, and the end of the light period as the product
of excess $^{13}$C (excess $^{13}$C in labeled sample versus relevant natural abundance control), core
volume, and concentration. Total excess flux of $^{13}$C as DIC or DOC (µmol $^{13}$C $m^{-2}$ $h^{-1}$) was
then calculated as:

7.  Excess $^{13}$C flux = (Excess $^{13}C_{start}$ – Excess $^{13}C_{end}$) / SA / $t$

where excess $^{13}C_{start}$ and excess $^{13}C_{end}$ represent excess $^{13}$C of DIC or DOC at the start and end
of dark and light incubation periods, SA is sediment surface area, and $t$ is incubation period
length (h). Net fluxes of excess $^{13}$C (excess $^{13}$C $m^{-2}$ $h^{-1}$) for DIC and DOC were calculated as:

8.  Net flux = ((dark flux * dark hours) + (light flux * light hours))/ 24 hours

Total $^{13}$C lost via flux to the water column from initial labeling to each sampling period was
interpolated from measured net flux values by calculating the area underneath the curve for



each treatment. To prevent the potential development of limitations during incubations, $O_2$
concentrations were not allowed to drop below 60% saturation in the dark and light
incubations were shortened (~3 h) to ensure that production was not limited due to available
resources within limited core volume.
**2.9 Data Analysis**

MPB-C biomass was determined using chl-a data for cores within all treatments for

0.5 d, 1.5 d, 2.5 d and 3.5 d. We therefore used a two-way analysis of variance (ANOVA) to
determine whether MPB-C biomass was affected by treatment and/or time. P/R ratios were
determined for 1.5 d, 2.5 d and 3.5 d to determine whether significant differences occurred
between treatments within each time period ($\alpha = 0.05$). Levene's tests indicated that variances
were homogeneous in all cases and there were no significant interactions between variables in
either analysis. For significant effects of treatment or time, post hoc Tukey tests were used to
identify significant differences between groups.

Total uptake for $^{13}$C into both MPB and bacteria, and relative $^{13}$C uptake into MPB

were determined for only a single core across all treatments from 0.5 d, 1.5 d, 2.5 d and 3.5 d.
To increase replication for statistical analysis, and therefore increase the power to detect a
significant difference, we therefore grouped data across times into two levels: before 1.5 d
(including 0.5 d and 1.5 d) and after 1.5 d (including 2.5 d and 3.5 d). There was no pooling of
data across treatments. A two-way ANOVA was then used to determine whether significant
differences occurred among treatments within each pooled time period ($\alpha = 0.05$). No
significant interactions were observed for total uptake into MPB or bacteria, but there was a
significant interaction observed for relative $^{13}$C uptake into MPB. For significant effects of



interaction, treatment, time, post hoc Tukey tests were used to identify significant differences
between groups.

The $^{13}$C remaining in microbial biomass and sediment OC were fitted with an

exponential decay function for each treatment across 3.5 d. Loss rate constants for these
exponential relationships were compared across treatments and are reported as positive
numbers following mathematical conventions associated with loss rates.
**3.0 Results**
**3.1 Uptake of nutrient additions**

Uptake of the added nutrients into the sediment was rapid and substantial, as indicated

by decreases in dissolved inorganic nitrogen ($NH_4^+ + NO_x$) concentrations in the overlying
core water to $<1.2 \pm 0.1$ µM L$^{-1}$ by 0.5 d. Across the incubation periods, elevated DIN
concentrations in overlying water were occasionally observed (Supplemental Fig. 1), but
corresponded with times when the cores were sealed for light and dark incubations, indicating
that DIN production was a result of in-core processing rather than nutrient amendments.
**3.2 Sediment characteristics**

Control sediment OC content was greater in the 2-5 cm depth ($187.5 \pm 27.7$ µmol C g$^{-1}$

$^{1}$) than at all other sediment depths ($112.3 \pm 11.4$ µmol C g$^{-1}$ in the TS, $149.8 \pm 31.6$ µmol C g$^{-1}$
$^{1}$ at 0-2 cm, and $120.1 \pm 16.5$ µmol C g$^{-1}$ at 5-10 cm). Natural abundance $\delta^{13}$C values were
most enriched in surface sediments (-18.7‰ in TS) and became progressively depleted within
deeper sediments to -22.1‰ at 5-10 cm (Table 1). In the 0-2 cm depth of the control sediment,



MPB-C biomass was 321.9 ± 42.0 mmol C m$^{-2}$ and bacterial biomass was 500.4 ± 65.3 mmol
C m$^{-2}$ (Table 1).

**3.3 Initial $^{13}$C uptake**

Uptake of $^{13}$C into sediment OC occurred rapidly and was observed in the first

cores collected (11 h after labeling, after tidal flushing, and with cores sectioned in the field).
At this time, prior to laboratory incubation and nutrient amendment, 1549 ± 140 µmol $^{13}$C m$^{-2}$
had been incorporated into sediment OC. Sediment OC was $^{13}$C-enriched across all sediment
depths at this time (Table 1), but 78% of the initially incorporated $^{13}$C was in the uppermost 2
cm of sediment (compared to 12.8% 2-5 cm, 9.4% 5-10 cm). Prior to incubation, $^{13}$C uptake
into microbial biomass at 0-2 cm was dominated by MPB (92.7 ± 1.6%), despite their lower biomass
(200.2 ± 26.5 mmol C m$^{-2}$) compared to bacteria (311.3 ± 56.4 mmol C m$^{-2}$) within the labeled cores.
Conversely, bacteria dominated $^{13}$C uptake in 2-5 cm sediment (66.8 ± 17.2% of the $^{13}$C
within microbial biomass). Although sediment OC at 5-10 cm was $^{13}$C-enriched, minimal
uptake was detected in microbial biomarkers.

**3.4 Effect of nutrient additions on P/R**

Average MPB biomass remained similar across treatments over the 3.5 d incubation

(two-way ANOVA: treatment $F_{3,31}$= 0.04, $p$=0.99; time $F_{3,31}$= 0.1, $p$=0.94, Supplemental
Figure 2). However, there were changes in P/R ratio that varied among treatments.
Examination of the effects of treatment and time on P/R showed no significant differences
(two-way ANOVA: treatment $F_{3,23}$=3.0, $p$=0.08; time $F_{2,23}$=2.7, $p$=0.11), although the post
hoc Tukey comparison between ambient and elevated treatments was nearly significant



(p=0.0506). For the ambient, minimal and moderate treatments, P/R ratios were dominated by
autotrophy and changed little over the first 2.5 d ($1.5 \pm 0.8$, $1.2 \pm 0.4$, and $1.3 \pm 0.1$,
respectively, Fig. 1b) as any increases in production were offset by increased respiration (Fig.
1a). By 3.5 d the minimal treatment had shifted into heterotrophy ($0.6 \pm 0.1$) as a result of
increased respiration and decreased production., whereas P/R ratios for the ambient and
moderate treatments remained essentially unchanged ($1.3 \pm 0.2$, $1.3 \pm 0.4$). P/R in the elevated
treatment was initially high compared to all other treatments ($2.2 \pm 0.2$ at 0.5 d) indicating
strong dominance of autotrophic production (Fig. 1a & B). However, P/R in the elevated
treatment generally decreased to $1.1 \pm 0.4$ after 3.5 d (Fig. 1), indicating a strong shift away
from autotrophy and towards dominance of heterotrophic processes as respiration increased
and production decreased (Fig. 1a).
**3.5 Incorporation of $^{13}$C into sediment organic carbon**
**3.5.1 Uptake of $^{13}$C into 0-2 cm sediment**
At 0.5 d, the $^{13}$C incorporated into sediment OC was predominantly contained in the 0-
2 cm depth across all treatments (~65%-90%, Fig. 2) and was statistically similar across
treatments (one-way ANOVA: $F_{3,7}= 4.2$, $p=0.1$). By 3.5 d, $^{13}$C retention was lower within
sediment from nutrient amended treatments compared to the ambient treatment. Whereas the
$^{13}$C contained in the 0-2 cm depth in the ambient treatment was similar across 3.5 d ($78.9 \pm$
8.8% 1.5 d, $77.0 \pm 16.4\%$ 2.5 d, $81.6 \pm 4.4\%$ 3.5 d), the $^{13}$C content decreased in the minimal,
moderate and elevated treatments to $70.3 \pm 8.3\%$, $73.6 \pm 16.4\%$, and $68.8 \pm 7.6\%$, respectively
(Fig. 2).



**3.5.2 Downward transport below 2 cm**


Downward transport of newly labeled material to 2-5 cm depth was low across all
treatments, but was higher for the elevated treatment at both 0.5 and 2.5 d. At 0.5 d there was
less downward transport in minimal and moderate treatments compared to the ambient and
elevated treatments. By 2.5 d downward transport was similar for ambient, minimal and
moderate treatments (10%, 9%, 10%, respectively; Fig. 2), but was considerably higher in the
elevated treatment (28.4%). By 3.5 d, $^{13}$C incorporation into 2-5 cm sediment OC was
similarly low for ambient, minimal and moderate treatments ($8.0 \pm 2.1\%$, $11.1 \pm 0.1\%$, and $8.7$
$\pm 2.1\%$, respectively), but lower in the elevated treatment ($4.8 \pm 2.1\%$). At 0.5 d, downward
transport into the 5-10 cm layer was a relatively small portion of initial $^{13}$C, but was higher in
ambient and minimal treatments ($8.7 \pm 2.4\%$ and $11.6 \pm 1.5\%$) when compared to moderate
and elevated treatments ($2.3 \pm 1.9\%$ and $6.8 \pm 0.1\%$, Fig. 2). Downward transport below 5 cm
was similar (5-11%) for all treatments at 2.5 d and 3.5 d.

**3.6 $^{13}$C distribution amongst sediment compartments**


**3.6.1 Microbial biomass**


The total $^{13}$C content of MPB (mmol $^{13}$C m$^{-2}$; Fig. 4a) decreased significantly from
before 1.5 d to after 1.5 d for all treatments (two-way ANOVA: $F_{1,8}=12.2$, $p=0.008$), but there
was no significant difference among treatments (two-way ANOVA: $F_{3,8}=2.7$, $p=0.12$). The
total $^{13}$C content of bacteria (mmol $^{13}$C m$^{-2}$; Fig. 4a) did not change significantly with time,
and was not significantly affected by treatment. The majority of the $^{13}$C assimilated into the
cores was present in the 0-2 cm depth (0-2 cm 2-5 cm $9 \pm 0.8\%$; and 5-10 cm $5.2 \pm 0.5\%$
Supplemental Fig. 3a, b & c). $^{13}$C incorporation was largely dominated by bacteria across all
treatments in sediments below 2 cm, with few exceptions. Increased bacterial contribution



occurred more quickly and was more pronounced in nutrient amended treatments at both 2-5
cm and 5-10 cm (Supplemental Figs. 4 & 5).

Total uptake of excess $^{13}$C (Fig. 4a), while informative about the amount of label

contained within each core at each time period, is not as useful for comparison between
microbial groups due to variations in the total amount of $^{13}$C assimilated between cores. It is
important to consider the relative contribution to $^{13}$C uptake (Fig. 4b) of both microbial groups
as each data point was sampled from separate cores that assimilated similar, but different,
initial concentrations of newly fixed $^{13}$C. Significant MPB contribution (%) decreased for after
1.5 d (two-way ANOVA: $F_{1,8}$= 83.1, $p$<0.0001) but showed no difference between treatments
( $F_{3,8}$=8.2, $p$=0.008), although interaction between the variables was significant ($F_{3,8}$=8.2,
$p$=0.008). Tukey tests found that MPB contributed less to microbial uptake of $^{13}$C in the
elevated treatment than in the ambient treatment ($p$=0.01) as well as the moderate treatment
being lower than the minimal treatment ($p$=0.014). MPB dominated the relative incorporation
of $^{13}$C into microbial biomass at 0-2 cm in all treatments initially (0.5d; 90% ambient, 90%
minimal, 92% moderate, and 92% elevated; Fig. 4b) and throughout the 3.5 d incubation (81-
90% ambient, 82-91% minimal, 74-92% moderate, and 65-92% elevated; Fig. 4b). The
relative bacterial contribution to microbial $^{13}$C incorporation increased across all treatments as
the incubations progressed, but increases in the moderate and elevated treatments at 2.5 and
3.5 d (Fig. 4b) corresponded with decreased $^{13}$C incorporation into MPB (Fig. 4a).
**3.6.2 Uncharacterized**

A portion of the $^{13}$C contained within sediment OC was uncharacterized, i.e., not

contained within the viable microbial biomass measured using PLFA biomarkers. Initially (0.5
d) the uncharacterized pool accounted for less sediment $^{13}$C within the nutrient-amended



treatments (1-3%) than within the ambient treatment (12%; Fig. 3), indicating that there was
more $^{13}$C contained in viable microbial biomass under increased nutrient availability after 12 h
of incubation. By 3.5 d increased contribution to the uncharacterized pool in the moderate and
elevated treatments (29% ambient, 32% minimal, 41% moderate and 45% elevated; Fig. 3)
corresponded with decreased $^{13}$C contained in MPB (52% ambient, 49% minimal, 42%
moderate and 26% elevated). In contrast, changes in the $^{13}$C in the uncharacterized pool did
not relate to $^{13}$C contained in bacteria, as the bacterial contribution to $^{13}$C remained relatively
unchanged (17% ambient, 14% minimal, 15% moderate and 15% elevated) and was similar
among treatments at 3.5 d.
**3.7 Loss of $^{13}$C from sediment OC**

Rates of $^{13}$C loss from sediment OC to the water column were highest in the moderate

and elevated treatments (total lost at 3.5 d: ambient 5%, minimal, 7%, moderate 11% and
elevated 20%; Fig. 5 & 6). Reflecting this, loss rate constants for the $^{13}$C remaining in
sediment OC after accounting for losses of DI$^{13}$C and DO$^{13}$C across 3.5 d were equivalent for
ambient and minimal treatments (0.018 ± 0.024, $R^2$ = 0.95 and 0.021 ± 0.001, $R^2$ = 0.99,
respectively; Fig. 6), but were higher for both moderate and elevated treatments (0.0383 ±
0.009, $R^2$ = 0.86, 0.0566 ± 0.003, $R^2$ = 0.99, respectively; Fig. 6).

Across all treatments, most of the $^{13}$C loss from sediment during the incubation

occurred via DIC fluxes (Fig. 5). Cumulative $^{13}$C export to the water column via DIC fluxes
was considerably larger than via DOC fluxes for all treatments (9× ambient, 11× minimal, 10×
moderate and 17× elevated). Initial DI$^{13}$C loss (0.5 d) was higher in the elevated treatment
than in the ambient, minimal, and moderate treatments (5.3 ± 3.4%, versus 0%, 1.1 ± 0.3%
and 1.4 ± 1.4%, respectively; Fig. 5). After 3.5 d, cumulative losses of DI$^{13}$C were higher in



moderate and elevated treatments (12.4 ± 11.6% moderate, 19.8 ± 10.8% elevated; Fig. 5)
than in ambient (4.0 ± 3.2%) and minimal treatments (6.6 ± 2.0%; Fig. 5 & 7).

DOC export was a less important pathway for $^{13}$C loss than DIC across all treatments.

$^{13}$C loss via DOC export was comparable and low across all treatments with similar maximum
export at 3.5 d (0.5 ± 0.2% ambient, 0.5 ± 0.2% minimal, 0.4 ± 0.2% moderate, and 0.6 ±
0.5% elevated; Fig. 5).

**4.0 Discussion**

This study examined the effects of enhanced nutrient loading on the processing

pathways for MPB-derived C in intertidal estuarine sediments. Enhanced nutrient availability
1) increased loss of MPB-derived C from sediment via DIC efflux (Fig. 5 & 6), 2) shifted
benthic metabolism to be less autotrophic (Fig. 1), and 3) decreased retention of C within
MPB (Fig. 3 & 4). These multiple lines of evidence indicate that intertidal sediments in areas
experiencing increased nutrient loading are likely to process C differently, resulting in reduced
potential for C retention within the sediment.

**4.1 Loss pathways for $^{13}$C**

Increased nutrient additions caused additional loss of $^{13}$C from sediment OC, largely

driven by DIC fluxes to the water column (Fig. 5 & 6). Complete loss of newly produced C
from sediment OC, as estimated from exponential decay functions, occurred more quickly in
nutrient amended treatments than in ambient (15% increase minimal, 210% increase moderate
and 310% increase elevated, Fig. 6). Increased loss rates indicated reduced turnover time for
newly produced MPB-derived C under increased nutrient load (419 d ambient versus 199 d
moderate and 134 d elevated). It should be noted that the loss rate constant for the minimal



treatment (0.021 ± 0.001, $R^2$ = 0.99, 366 d) was comparable to that for the ambient treatment
(0.018 ± 0.024, $R^2$ = 0.95, 419 d), indicating that a small nutrient addition may not cause
significant decreases in C turnover time. Increased loss rates imply that C retention and burial
in MPB-dominated photic sediments are greatest when nutrients are limiting and that
increased nutrient availability alters the processing of MPB-C within the sediment. Increased
nitrogen availability appears to have decreased the retention of C within MPB biomass (Fig.
3). Increased turnover of the newly fixed MPB-derived C from the sediment likely occurred as
the net result of exudation of material and breakdown of cells. This increased turnover may
have caused the increased efflux of MPB-derived C as exudates and cell components were
increasingly available to support respiration.

Across all treatments, DIC was the main loss pathway for MPB-C, DOC was a minor

pathway and loss via $CO_2$ was considered negligible (Oakes and Eyre 2014) (Fig. 5 & 6). Loss
of $^{13}C$ via the DIC pathway appears to be stimulated by nutrient additions, resulting in
increased export occurring earlier within incubations as a result of increased bacterial
remineralization (Fig. 2 & 5). Increased DI$^{13}$C export represents the portion of DI$^{13}$C
produced via respiration in excess of that which is re-captured and utilized by MPB to drive
production. Given the close proximity of bacteria and MPB in the sediment, there is the
potential for considerable utilization of the DI$^{13}$C arising from bacterial remineralization to
support algal production. Relatively low fluxes of DI$^{13}$C to the water column in the ambient
treatment across 2.5 d likely indicate more complete utilization and recycling of DI$^{13}$C to
support algal production (Fig. 5). Export of DI$^{13}$C was considerably higher in both the
moderate and elevated treatments, indicating production of DI$^{13}$C during bacterial
remineralization in excess of utilization of DI$^{13}$C by MPB. Decreased recycling of DI$^{13}$C from



remineralization in elevated treatments could develop due to 1) decreased DIC demand as
algal production decreased after initial stimulation or 2) increased production of unlabeled
DIC through remineralization of previously refractory organic material providing an
alternative unlabeled source to support algal production.
Cumulative losses of $DO^{13}C$ were low for all treatments across 3.5 d (<1.5 % of total
$^{13}C$, Fig.5) and did not appear to change significantly with increased nutrient availability.
Previous studies have also found that DOC fluxes are a relatively minor contributor to loss of
MPB-derived carbon (Oakes et al. 2012; Oakes and Eyre 2014), as observed in the current
study, but DOC may be a significant export pathway in other settings. Produced DOC may be
labile and respired to DIC prior to loss from the sediment, but this pathway was not greatly
altered in this study due to increased nutrient availability.
**4.2 Shifts in benthic metabolism**
Each nutrient amendment produced a different shift in benthic metabolism within the
core incubations (Fig. 1) with no clear dose-effect relationship between increased nutrient
availability and P/R observed among nutrient-amended treatments. Heterogeneity in both
bacterial and MPB biomass are routinely observed within intertidal sediment and can lead to
substantial variability between the production and respiration observed between cores (Eyre et
al. 2005; Glud 2008). Despite a background of variability between cores, both minimal and
elevated treatments display a decrease in autotrophy. The minimal treatment shifted into
heterotrophy (P/R<1) and the elevated treatment stimulated initial algal production sufficient
to cause a subsequent spike in respiration. Increased respiration by 3.5 d was partially offset
by maintained production that kept P/R above 1. In contrast, the moderate treatment
maintained a steady P/R across 3.5 d, although substantial error bars indicate considerable



variability between the cores within the treatment. Differences in the response among nutrient-
amended treatments appear to result from increased initial production that was supported in
both the elevated and moderate treatments, but that decreased by 3.5 d in the minimal
treatment. MPB-dominated sediment is expected to be net autotrophic, with positive GPP
(Tang and Kristensen 2007) that may be further stimulated by nutrient inputs (Underwood and
Kromkamp 1999). Increased algal production of labile organic matter subsequently stimulates
heterotrophic respiration, increasing oxygen consumption and lowering P/R (Glud 2008;
McGlathery et al. 2007). Quick increases in MPB productivity followed by increased
respiration have been observed in response to pulses of organic matter in both oligotrophic
and estuarine sediments (Eyre and Ferguson 2005; Glud et al. 2008). Rapid increases in
respiration rates, as reflected in the oxygen fluxes for the elevated treatment (Fig. 1a), are
often associated with an increased supply of labile C and can occur at rates higher than
expected for in situ temperature. This has been observed in subtropical sediments (Eyre and
Ferguson 2005) as well as polar and temperate systems (Banta et al. 1995; Rysgaard et al.
1998). Although the sediments in this study were not oligotrophic, the extent of the shift
towards heterotrophy is still likely controlled by the amount and relative quality (C/N ratio) of
the organic matter available for processing (Cook et al. 2009; Eyre et al. 2008). The similarity
in initial P/R ratios between ambient and minimal treatments indicate that a small nutrient
addition did not stimulate large increases in algal production, but rather a small increase in
production that was offset by increased respiration in the minimal treatment (Fig. 1). The
moderate treatment had a distinctly different reaction to increased nutrient availability, with
stable P/R as both production and respiration were maintained across 3.5 d. The elevated
treatment had increased algal production at 1.5 d, with the highest production rate observed in

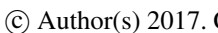



this study, and this was followed by a considerable increase in respiration by 3.5 d (increased
dark uptake of $O_2$; Fig. 1). Decreased autotrophy by 3.5 d was a result of both elevated
respiration driven by increased bacterial decomposition of labile material and declining MPB
production. It is important to note that the elevated treatment did not shift to a P/R less than 1,
but did display a considerable increase in respiration. The rapid increase in respiration in the
elevated treatment suggests that the newly produced organic matter was readily bioavailable
and quickly processed by bacteria as a result of increased nutrient availability.
**4.3 Retention of carbon within microphytobenthos biomass**

Within surface sediments, MPB biomass did not increase with increased nutrient load,

despite apparent increases in productivity (Supplemental Fig. 2). Although MPB biomass did
not change, by 3.5 d the $^{13}$C retained within MPB biomass in the nutrient-amended treatments
appears to have decreased (Fig. 4a) indicating increased turnover of newly fixed C out of
MPB biomass. This aligns with many previous reports that increased productivity does not
necessarily correspond with increased algal biomass (Alsterberg et al. 2012; Ferguson and
Eyre 2013; Ferguson et al. 2007; Hillebrand and Kahlert 2002; Piehler et al. 2010; Spivak and
Ossolinski 2016). Lack of change in MPB biomass, despite increased productivity, may occur
as a result of grazing or secondary nutrient limitation (Hillebrand and Kahlert 2002), but these
explanations are unlikely for the current study. Grazing is likely to have occurred at only a
low level. There was very little fauna, including grazers, within sediment at the study site and,
although any grazers such as copepods that were within the site water would have been
included in the incubations, larger, mobile grazers were excluded. Secondary nutrient
limitation of P or Si was avoided through additions of both elements at 0 d for P and 2.5 d for
Si during incubation. It is more likely that the microbial community responded to pulses of





increased nutrients through increased production of extracellular compounds (MPB:
carbohydrates; bacteria: enzymes) rather than increasing their biomass (Thornton et al. 2010).
This may be a strategy to optimally utilize intermittently available nutrient resources, given
that increased cell numbers (biomass) within a biofilm community may otherwise increase
competition among cells (Decho 2000; Drescher et al. 2014). Allocation of additional N
towards increased production of extracellular enzymes or storage molecules rather than new
biomass may therefore benefit the community. Strong competition between MPB and bacteria
for available N resulted in a minimal contribution from denitrification as a pathway for N loss
likely as a result of limited availability of $NO_3^-$ for denitrifying bacteria (unpubl. data).
**4.4 $^{13}$C distribution within the sediment**
**4.4.1 Microbial biomass**

Decreased autotrophy is somewhat reflected in the relative partitioning of $^{13}$C from

newly produced algal C between MPB and bacteria within the individual treatments (Fig. 1 &
Fig. 4b). Initially, uptake of $^{13}$C was strongly dominated by MPB amongst treatments, with
minimal incorporation by bacteria. As incubations progressed, a shift towards increased
relative contribution by bacteria was apparent in all treatments, but was more substantial in the
elevated treatment (3.5 d; 19% ambient, 18% minimal, 26% moderate, and 35% elevated, Fig.
4b). This quicker shift towards bacterial dominance of $^{13}$C incorporation corresponded with
the largest decrease in P/R ratios observed in this study, as increased respiration and decreased
production caused the elevated treatment to become less autotrophic (Fig. 1). These
corresponding factors are likely a result of a tight coupling and intense recycling between
algal production and bacterial processing of newly produced MPB-derived C. EPS can be a
large export pathway for newly fixed C from algal cells (up to 70.3% Goto et al. 1999) and





can provide a labile C source for heterotrophic or denitrifying bacteria. The $^{13}$C incorporated
into bacteria represents the balance of respiration and uptake and is expected to become
increasingly muddled by $^{13}$C being processed through other pathways (denitrification) as
incubations progress. Therefore, this study only considered the transfer of MPB-C into
bacteria at the 0.5 d sampling. However, given the low initial transfer of $^{13}$C to bacteria in all
treatments over 24 h following labeling (0.5 d; 0.8% h$^{-1}$ ambient, 0.8% h$^{-1}$ minimal, 0.7% h$^{-1}$
moderate, 0.7% h$^{-1}$ elevated; Fig. 3) it appears that either production or utilization of EPS
containing newly fixed C was relatively low in the current study, regardless of nutrient
addition. This transfer was the net result of EPS production and bacterial remineralization and
would have become increasingly muddled as $^{13}$C-containing detrital material accumulated as
incubations progressed. Low EPS production at 0.5 d may indicate that N is not limiting for
MPB in these sediments, as exuded EPS does not appear to be copious, as would be expected
under severe N limitation (van Den Meersche et al. 2004). Similarly low rates of C transfer
from MPB to bacteria were previously reported for the site (0.83% h$^{-1}$, Oakes and Eyre 2014)
and are towards the lower end of the range of EPS production rates for benthic diatoms (0.05
to 73% h$^{-1}$; Underwood and Paterson 2003). At 0.5 d nutrient availability appears to have had
little effect on the initial transfer rates from MPB to bacteria, but appears to have decreased
the turnover of MPB-C out of the microbial community, as contributions of $^{13}$C to the
uncharacterized pool were lower in the nutrient-amended treatments (Fig. 3). By 3.5 d,
increased nutrient availability appears to stimulate the transfer of $^{13}$C from microbial biomass
in the uncharacterized pool, but had no effect on $^{13}$C in bacteria as the bacterial pool was equal
across all treatments (15-18%, Fig. 3 & 7).



### 4.4.2 Uncharacterized


A portion of the $^{13}$C incorporated into sediment OC was uncharacterized (i.e., not
within microbial biomass). By 3.5 d, the portion of initially incorporated $^{13}$C that was within
the uncharacterized pool varied substantially among the treatments (29-46%, Fig. 7). This
uncharacterized C is likely to represent a mixture of both labile and refractory OC (Veuger et
al. 2012), including metabolic byproducts, senescent cells undergoing breakdown, EPS,
extracellular enzymes, carbohydrates, and a variety of complex, molecularly uncharacterized
organic matter (Hedges et al. 2000). Collectively, these molecules form a pool of labeled intra
and extra-cellular material remaining in sediment OC derived from both MPB and bacteria
that is not characterized as microbial biomass when using PLFAs to estimate microbial
biomass (e.g., $^{13}$C contained in storage products or enzymes that was not incorporated into
phospholipids). Given that MPB can direct up to 70% of their newly fixed C to EPS (Goto et
al. 1999), carbohydrates are likely to form a considerable portion of the uncharacterized $^{13}$C.
A study using a similar $^{13}$C-labeling approach reported that 15-30% of MPB-derived carbon
was transferred to intra- and extracellular carbohydrates within 30 d after an initial transfer
rate of ~0.4% into bacteria (2 d; Oakes et al. 2010a). In light of the higher transfer rates for
$^{13}$C into bacteria observed in this study (0.7 to 0.9% h$^{-1}$), there is potential for a considerable
portion of the uncharacterized pool to be accounted for by EPS.
When quantified, the uncharacterized C pool typically has a high C:N ratio (10 to 60;
Cook et al. 2009; Eyre et al. 2016a), indicating that nitrogen availability may have a role in
regulating its content and accumulation. Given that nitrogen limitation has been observed to
suppress processing pathways of otherwise labile OM in soils (Jian et al. 2016; Schimel and
Bennett 2004), a similar mechanism may be possible in estuarine sediments. This mechanism



may include a priming effect due to either increased production of extracellular enzymes or
due to increased energy from labile C compounds allowing for the increased breakdown of
sediment OM (Bianchi 2011). Increased extracellular enzyme production would result in more
complete utilization of sediment OM through promotion of hydrolysis (Arnosti 2011; Huettel
et al. 2014), a potentially limiting step during the breakdown of organic material. This would
result in more complete utilization of $^{13}$C by microbial biomass and a smaller pool of
uncharacterized C within sediment OC, as was observed in the nutrient-amended treatments at
0.5 d (Fig. 3). This is further supported by the increased turnover of MPB-C from microbial
biomass into the uncharacterized pool observed within the nutrient amended treatments (2.5 d,
Fig. 3) indicating $^{13}$C that was previously incorporated into MPB was processed into the
uncharacterized pool more quickly with increased nutrient availability. After 2.5 d, the $^{13}$C
content of the uncharacterized pool was substantially larger for the elevated treatment (Fig. 3
& 7) and looks to have been largely sourced from MPB $^{13}$C, given that bacterial contribution
to sediment OM remained stable. Composition of the uncharacterized pool will be study-
specific depending on the different biomarker techniques utilized to estimate microbial
biomass incorporating different pools of material. The metabolic pathways and ecological
strategies regulating the portion of $^{13}$C entering the uncharacterized pool warrant further
investigation.
**4.5 Downward transport**

Increased nutrient availability reduced the downward transport of fixed $^{13}$C,

particularly within 2-5 cm, mainly as a result of increased export of MPB-C to the water
column. In the ambient treatment, downward transport to 2-5 cm (10.0%) and 5-10 cm (9.2%)
across 60 h was comparable to that reported by Oakes and Eyre (2014) for the same site (8.3%



2-5 cm, 14.9% 5-10 cm, 60 h). Oakes and Eyre (2014) suggested that resuspension resulting
from a flood event limited the downward transport of $^{13}$C, but a comparable and lower rate of
downward transport at 60 h (12.1% 2-5 cm, 9% 5-10 cm, ambient treatment) was observed in
the current study in the absence of marked freshwater inflow. Downward transport is not a
large pathway for loss of $^{13}$C within this system as transport to sediment below 2 cm was
minimal, and appeared further reduced in the elevated treatment (Supplemental Fig. 3b & c).
Decreased downward transport of MPB-derived C under increased nutrient load may reflect 1)
decreased transport to depth as diatoms reduce migration downward to find nutrients
(Saburova and Polikarpov 2003) or 2) relaxation of the tight recycling and retention of newly
fixed C between MPB and bacteria within surface sediments allowing for increased export of
labile C to the water column (Cook et al. 2007). Decreased downward transport in this study
likely reflects a combination of reduced algal transport of $^{13}$C to depth and increased loss of
$^{13}$C from surface sediments to the water column.
**4.6 Implications**

This study has provided valuable insight into the processing of MPB-derived C under

increased nutrient availability using multiple lines of evidence (budgeting $^{13}$C within sediment
compartments and sediment-water effluxes, partitioning of C pools via biomarkers, and
changes in P/R) and is among the first to have addressed this problem. However, some caveats
on interpretation are important to note, as follows: 1) *Ex situ* incubation of sediment cores
may not be directly comparable to processes occurring *in situ* and may overestimate C
retention, as there is reduced potential for loss via sediment resuspension due to tidal
movement, water currents, and grazing. 2) Removal of grazers may also increase MPB
production and their release of exudates (Fouilland et al. 2014), which could enhance $^{13}$C





transfer to bacteria. However, given the lack of apparent grazers at the site of the current
study, and the low observed $^{13}$C transfer rate to bacteria (0.7-0.9% h$^{-1}$ Fig. 4b) that was
comparable to previously measured in situ rates in Oakes and Eyre (2014), grazers appear to
have had little potential impact on sediment processing in this study.

The findings show that increased nutrient availability reduced C retention, but the

main export pathway for algal carbon remained the same (primarily loss via DIC). Coastal
environments are recognized as important sites for carbon storage. Although the focus has
primarily been on vegetated environments (Duarte et al. 2005), which store the most carbon,
unvegetated sediments also have capacity for longer-term retention (e.g. 30% after 30 d Oakes
and Eyre 2014; 31% after 30 d Oakes et al. 2012). Based on N burial rates (and corresponding
unpublished C burial rates) some coastal systems can have higher C burial rates in subtidal
and intertidal macrophyte-free MPB sediments than in macrophyte-dominated sediments
(Eyre et al. 2016b; Maher and Eyre 2011) although this was shown in only one of the three
estuaries studied. Increased nutrient loading into coastal settings has been implicated in
historical decreases of long-term carbon storage through a shift from macrophyte dominated
systems (seagrass and mangrove) towards MPB dominated systems (Macreadie et al. 2012)
within coastal environments. Carbon storage potential within MPB dominated sediments
remains a significant knowledge gap within the carbon budgets of estuaries. At 30 d, estimates
of retention of C identified for ambient and minimal treatments were considerable in the
current study (58% and 54%), however, increased nutrient loading reduced this retention
considerably (32% moderate, 18% elevated). Given that nutrient inputs have increased
globally and bare photic sediment accounts for a large surface area within estuaries, these two
factors could have resulted in substantial release of currently stored carbon and demonstrate



the capacity for further substantial reduction of C storage potential globally if elevated
nutrient inputs continue within estuarine systems.

Although MPB-dominated sediments probably have less decadal-scale long-term

storage of C than macrophyte-dominated sediments, this study clearly demonstrates that the
existing storage potential is further degraded by increased nutrient loading within MPB-
dominated sediments. These sediments may lock away less C per area, but are fairly
ubiquitous within photic coastal and oceanic sediment and may contribute significantly to
carbon storage within coastal systems due to this increased area. The observed increases in
mobility of newly fixed algal carbon from intertidal sediments (Fig. 5) as a result of elevated
anthropogenic nutrient loading will directly translate to increased carbon export to coastal
oceans and reduced carbon storage potential within shallow photic estuarine sediments.

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

**Author Contribution**
PR planned the experimental design and field work, performed the field work, isolation of
biomarkers and laboratory analysis, and wrote the manuscript. JO planned the experimental



design and field work, contributed to the data interpretation and assisted with statistical
analysis and writing of the manuscript. BE planned the experimental design and field work,
contributed to the data interpretation and assisted with the writing of the manuscript. The
group of co-authors has approved the submission of this manuscript.
**Acknowledgements**

We thank Natasha Carlson-Perret and Jessica Riekenberg for field assistance, Iain

Alexander for laboratory analysis, and Matheus Carvalho for isotope analysis. This study was
funded by an Australian Research Council (ARC) Linkage Infrastructure, Equipment and
Facilities grant to B.D.E. (LE0668495), an ARC Discovery Early Career Researcher Award to
J.M.O. (DE120101290), an ARC Discovery grant to B.D.E. (DP160100248), and ARC
Linkage Grants to B.D.E. (LP110200975; LP150100451; LP150100519).



**Figures and Tables**

| | | Top Scrape | | | | 0 to 2 cm | | | | 2 to 5 cm | | | | 5 to 10 cm | | | |
|---|---|---|---|---|---|---|---|---|---|---|---|---|---|---|---|---|---|
| | | Biomass | SE | $\delta^{13}$C | SE | Biomass | SE | $\delta^{13}$C | SE | Biomass | SE | $\delta^{13}$C | SE | Biomass | SE | $\delta^{13}$C | SE |
| **Control cores** | Sediment organic carbon | 318.0 | 32.3 | -18.7 | 0.3 | 3818.0 | 804.2 | -20.7 | 0.3 | 7963.3 | 1174.9 | -22.0 | 0.4 | 8498.2 | 1165.2 | -22.1 | 0.4 |
| | Microphytobenthos biomass | | | | | 321.9 | 42.0 | | | 226.2 | 33.1 | | | 227.3 | 37.3 | | |
| | Bacterial biomass | | | | | 500.4 | 65.3 | | | 286.0 | 68.8 | | | 244.1 | 66.1 | | |
| **Initial cores** | Sediment organic carbon | 376.4 | 4.5 | 121.4 | 23.7 | 3693.6 | 382.4 | -7.5 | 2.1 | 5056.8 | 117.8 | -19.4 | 1.0 | 8397.0 | 492.5 | -21.4 | 0.7 |


**Table 1:** $\delta^{13}$C values (‰) and carbon biomass (μmol C m$^{-2}$) for control (natural abundance,
n=3) and initially labeled cores (n=3, 0 d). Microphytobenthos and bacterial biomass are only
provided for control cores.

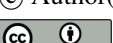


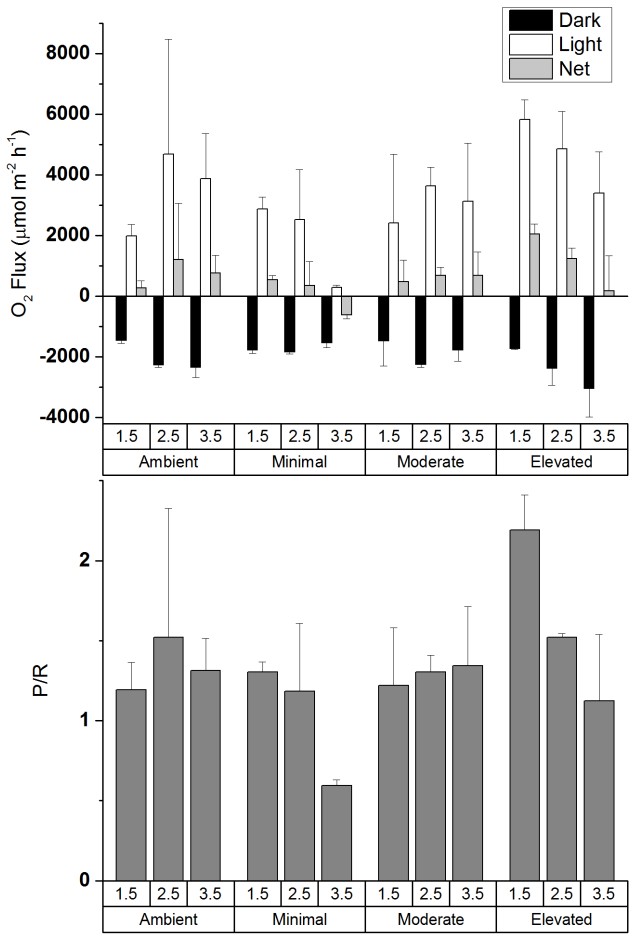


**Figure 1:** Oxygen fluxes and ratio of production to respiration (P/R) for all treatments across

24 h calculated from oxygen fluxes for individual cores. Values are mean ± SE.





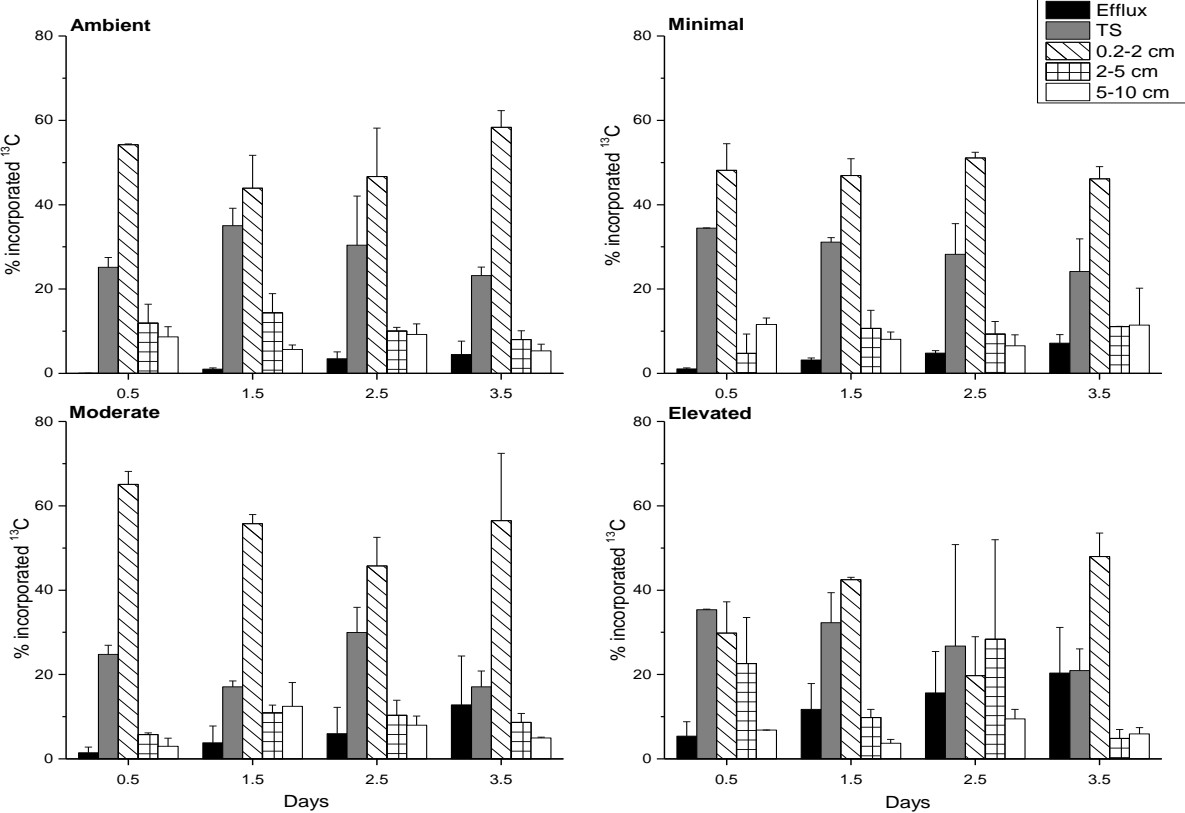

933

**Figure 2:** Carbon budget for excess [13]C within sediment OC at top scrape (TS), 0.2 to 2 cm, 2

to 5 cm, 5-10 cm, and the cumulative excess [13]C exported to the water column via the

combined efflux of DIC and DOC for each treatment at each sampling time. All values are as

a percentage of the [13]C initially incorporated into sediment OC (0-10 cm). Some error bars are

too small to be seen (mean ± SE).






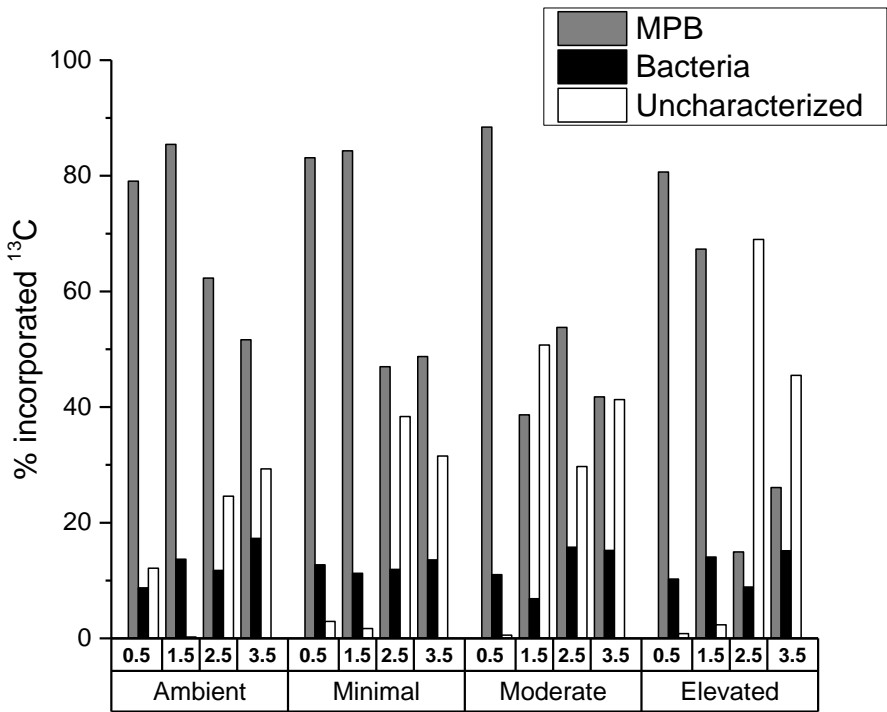


**Figure 3:** Excess $^{13}$C incorporation into microphytobenthos, bacteria, and uncharacterized OC

as a percentage of $^{13}$C contained in sediment OC in 0-10 cm. There are no error bars as PLFAs
were analyzed for only one replicate sample from each time period.

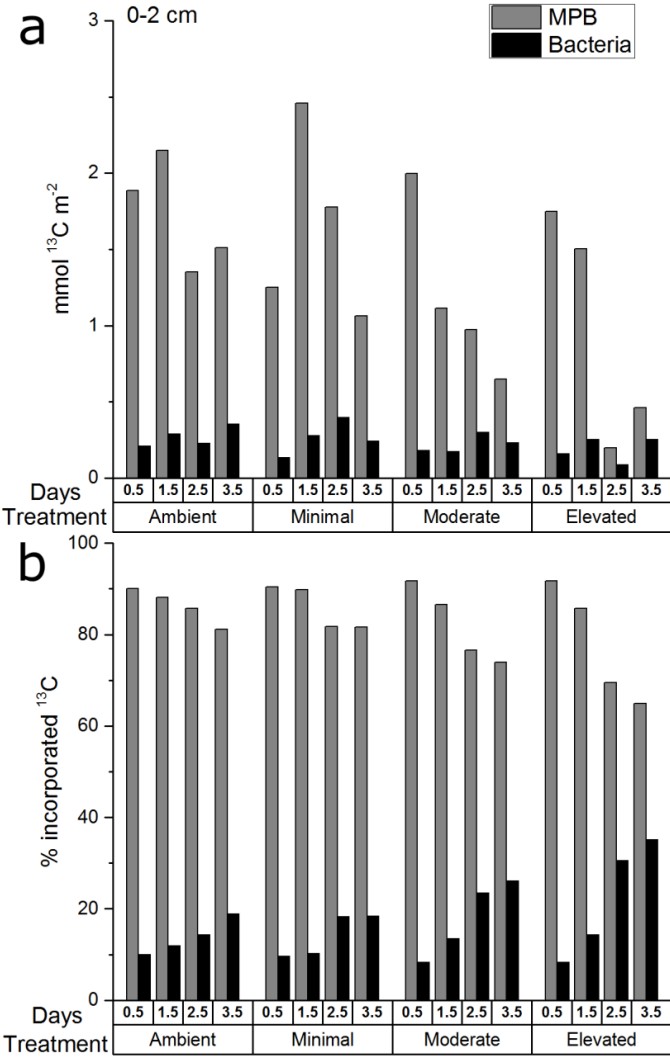

94

**Figure 4:** [13]C within MPB and bacterial biomass in sediment at 0-2 cm depth as A) total

excess [13]C (mmol [13]C m[-2]) and B) a percentage of the total [13]C in microbial biomass at 0-2 cm

at each time period. There are no error bars as PLFAs were analyzed for only one replicate

sample from each time period.




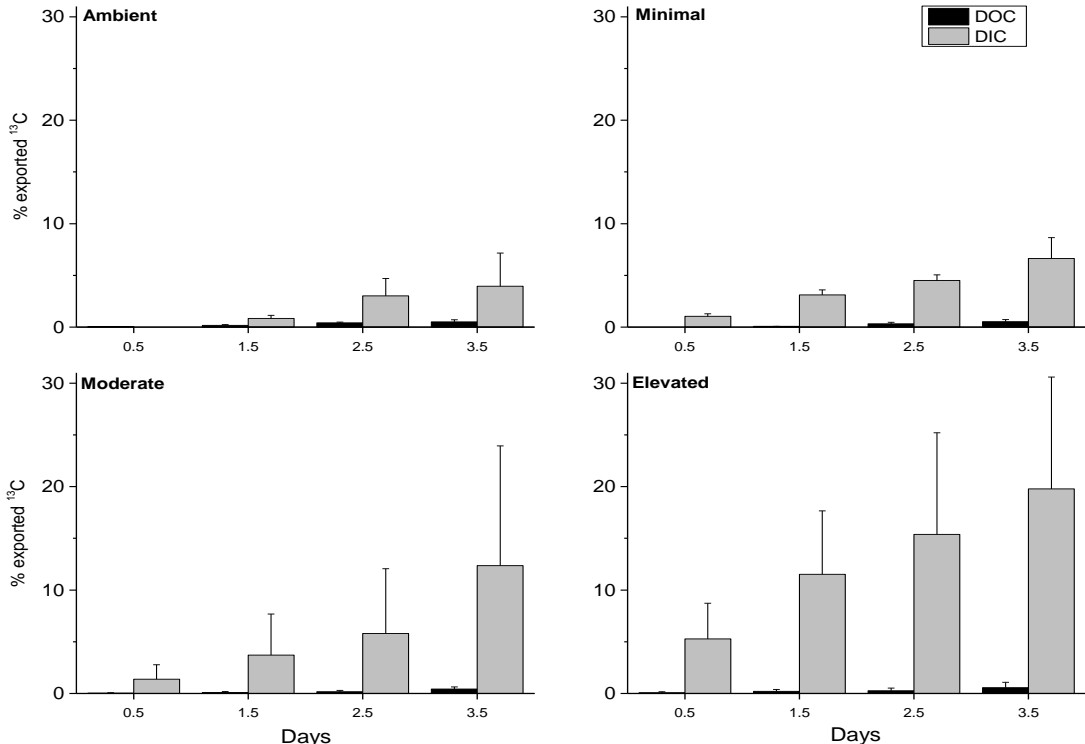


**Figure 5**: Effluxes of $^{13}$C from the sediment as dissolved organic carbon (DOC) and dissolved

inorganic carbon (DIC) as a percentage of the total $^{13}$C contained in sediment at 0-10 cm

depth at each sampling time (mean ± SE).





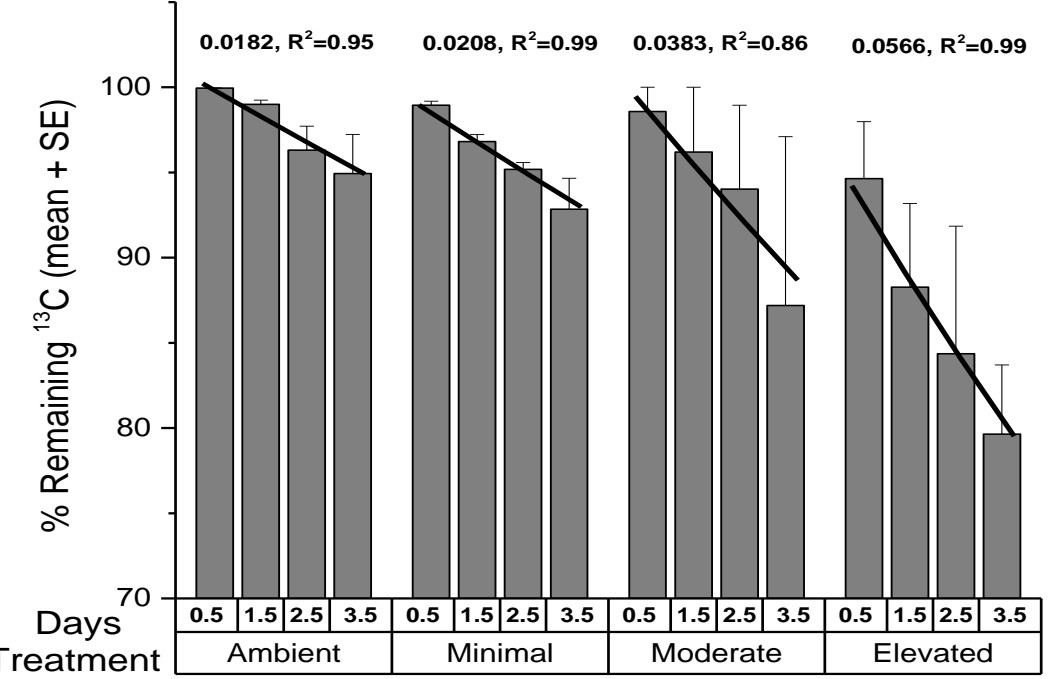


**Figure 6**: The percentage of $^{13}C$ remaining in sediment OC (0-10 cm depth) after accounting

for losses of $DI^{13}C$ and $DO^{13}C$ to the water column (mean ± SE). Lines are exponential decay

functions for each treatment across the 3.5 d of incubation (Loss rate constant, $R^2$ of function).



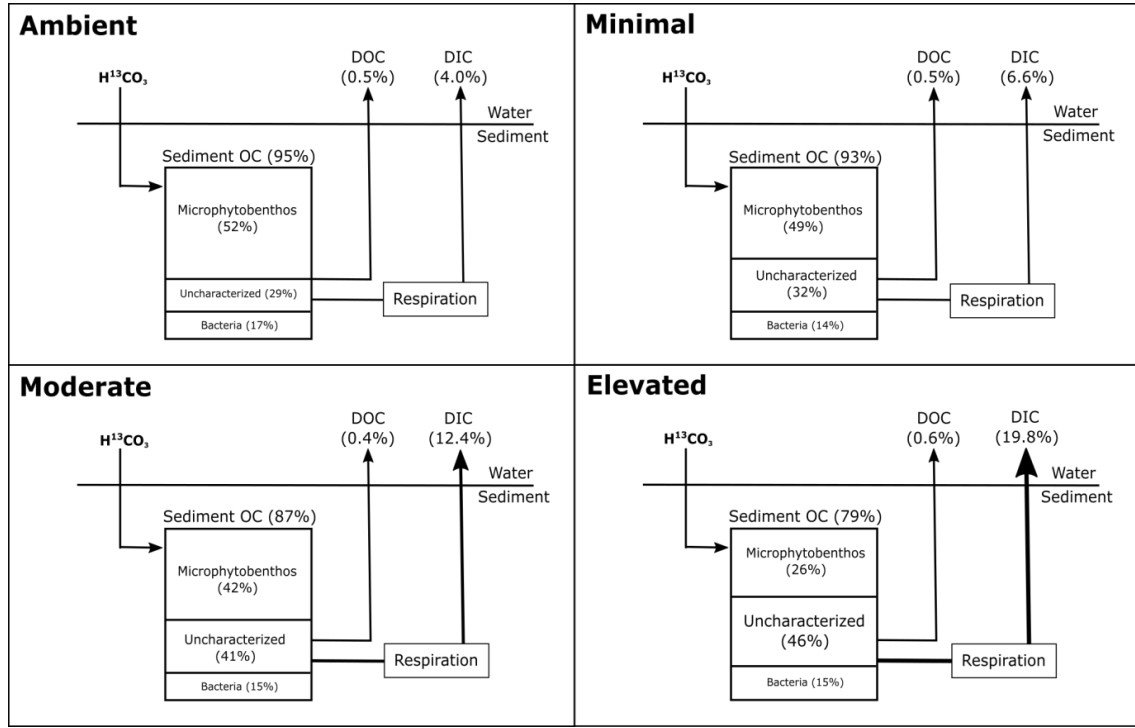


**Figure 7:** Distribution of $^{13}$C at 3.5 d of incubation of inundated sediment including loss

pathways for DIC and DOC. The $^{13}$C contained in sediment organic carbon (sediment OC) is

further partitioned into microphytobenthos, bacteria, and uncharacterized organic carbon as a

percentage of the $^{13}$C in sediment organic carbon at 0-10 cm 3.5 d after labeling (Figure layout

from Eyre et al., 2016).