# Peer review of "Short-term fate of intertidal microphytobenthos carbon under enhanced nutrient"

_Biogeosciences, 2017_

## Referee Comment (RC1) · Anonymous Referee #1 · 18 Dec 2017

General comments

The goal of this study was to quantify the effects of nutrient additions on carbon transformations in coastal sediments dominated by MPB and bacteria. Understanding the effects of nutrient loading on benthic microbes is important because coastal ecosystems are increasingly impacted by eutrophication and MPB are the dominant primary producer in unvegetated, shallow sediments. Consequently the authors are addressing questions that are likely of interest to readers of Biogeosciences. Although the research questions are interesting and timely, there are several issues that the authors should address.

The authors applied a 13C label to exposed surface sediments at low tide and then waited 11 h to collect cores (lines 128-130, 143-144). Approximately 2 h after collection, the cores were placed in the experimental tanks and exposed to the nutrient treatments (lines 150, 155-161). Carbon exchange, recycling, and loss between MPB and bacteria occurs quickly – over just a few hours. My concern is that at least 13 h passed between the 13C label addition and the application of the nutrient treatments. It seems that carbon exchange between MPB and bacteria that occurred prior to nutrient additions would confound the effects of the nutrient treatments. In addition, it is not clear from S1 that the nutrient treatments were effective. Further, it seems from the methods that the cores in the same nutrient treatment tank all shared the same water column (lines 171-173). This would affect the independence of processes across cores and could confound the results.

For the incubations, cores were sealed for 30 mins before initial samples were collected. Final samples were collected 3 h (light), 12 h (dark), or up to 16 h later (lines 182, 184-185). It would be good to clarify the exact duration of light and dark incubations and why the times differed between conditions. The authors should also clarify whether rates were calculated based on concentration changes between two time points. Generally, at least 3 time points are needed to calculate fluxes and capture non-linear dynamics. Moreover, long incubations, particularly under dark conditions, likely created 'bottle' effects that could affect metabolic rates. The authors should discuss potential artifacts of the sampling approach – particularly the potential impact of low DO on sediment respiration rates. Additional rationale for the time course of measuring fluxes 6 h after adding nutrients (and 19 h after adding 13C), and then again 1.5 d, 2.5 d, and 3.5 d would also be useful.

The authors need to provide more rationale for using 16:1n7 as a marker for diatoms. While this compound is an important component of diatom lipids, it is also produced by other groups of algae as well as by iron reducing bacteria and sulfate reducing bacteria. These communities are likely active in shallow coastal sediments and could be

in close proximity to MPB if the sediments are anoxic. The absence of polyunsaturated C20 and C22 PLFAs seems to suggest that diatoms and other microalgae were at low abundances. I am not convinced that the current approach adequately isolates contributions from diatoms vs. non-diatoms.

The carbon mass balance calculations should be clarified. For instance, the authors report the %C loss from the sediments, but it is unclear how this was calculated. Did this reflect the % change between the initial core collections and each sampling time point (0.5, 1.5, 2.5, and 3.5d)? Or was this calculated by subtracting out 13C losses via DIC and DOC (lines 446-448)? If the former, then that should automatically account for losses by respiration and exudation. Similarly it is unclear how the exponential decay functions were calculated. The error bars in Figures 5 and 6 are large, particularly for the moderate and elevated treatments; this variability makes the substantial differences across treatments (lines 473-476) somewhat unexpected.

The discussion should be more concise and focused. It would have been helpful if the authors considered processes driving variability across the treatments and compared their findings to other studies examining MPB-bacterial responses to surface water nutrient additions – particularly other stable isotope labeling experiments. Along these lines, I was somewhat surprised that nutrient additions would not stimulate MPB production and thereby promote C retention in the sediments.

Specific comments Introduction: The authors acknowledge that other studies have examined the effects of nutrient loading on MPB (lines 73-77), but should discuss the subset of studies that used isotope tracer techniques in more depth, as these are directly relevant to the current manuscript.

Line 113 and elsewhere: instead of reporting carbon concentration per surface area, please report % organic carbon.

Lines 155-160. What were the target concentrations of each of the treatments and was the site water filtered before the treatments were applied?

Calculations: The biomass calculations use several conversion factors to scale from lipid concentrations to bacterial and diatom biomass. It is unclear how well constrained the conversion factors are and the amount of uncertainty they introduce into the calculations. For instance, does the concentration of lipid per unit biomass change with algal growth or nutrient condition? It is unclear why the lipid concentrations need to be scaled to biomass – why not compare the ratio of bacterial to microalgal lipids?

Equations 3-4 are descriptions more so than equations.

Line 309: ANOVA is not the most appropriate test because the cores within a treatment were not independent of one another as they shared a water column.

Line 318: What is the ecological rationale for grouping time points 0.5 and 1.5 vs. 2.5 and 3.5?

Lines 388-400. How did downward transport occur in these cores? MPB are generally restricted to the top 2 cm and it does not seem that there was pore water flow during the lab incubations. Was there mixing by animal communities? Is it possible that contamination occurred during core collection in the field?

Line 431: Was the uncharacterized fraction defined as PLFAs that were not i,a-15 or 16:1n7? If so, additional rationale is needed to justify this approach.

Graphs: It would be helpful if the error bars were positive and negative
* * *

---

## Referee Comment (RC2) · Anonymous Referee #2 · 10 Jan 2018

The manuscript by Riekenberg et al. describes data from 13C-incubation experiments whereby microphyobenthos was labeled with 13C in situ, and then incubated under controlled conditions over a period of 3.5 days; either with background nutrient levels or with higher than ambient N &/or P concentrations.

While generally a well performed study, I am surprised by the short duration of the experiments (3.5 days). When fitting exponential decay functions on the resulting data (Figure 6), I feel this is somewhat thin ice – the data should be spread more in time for a convincing exponential fit.

-Abstract, line 30 and in Discussion: clearly define in the manuscript how you define

and calculate the turnover time, to avoid any ambiguity. I find these turnover times surprisingly high (i.e., long), and in line with other comments, wonder whether the short incubation period did not lead to a bias in this estimate – with 3 time points very early on it seems not ideal to fit an exponential fit to these data. Also, it is not unambiguously clear what your t=0 is (after the 6 hour 'acclimation period' ? See next comment).

On page 9, line 179, the authors mention that the cores were allowed to 'acclimate for 6 hrs prior to the start of the incubation'. I'm not sure what this means, it's not as if no microbial activity would take place during this period, hence for me it would seem to be an integral part of the incubation period. Why not simply define t=0 as the moment the cores were no longer exposed to 13C-DIC labeling ? Are these 6 hours part of the incubation times mentioned throughout the ms ? If not, this may bias the estimates of turnover times.

In the abstract (line 26-27), the authors mention that treatments with higher nutrient levels showed higher loss of 13C label, "supporting increased production of extracellular enzymes and storage products". I have two reservations here: First, this pattern would equally be consistent with a scenario in which the heterotrophic bacterial community was N and/or P-limited ? Eg Keuskamp et al. Sci Total Environ. 2015 doi: 10.1016/j.scitotenv.2014.11.092. I would suggest to add this as a possible mechanism in the intruduction on page 4 (section starting at line 68). Secondly, this conclusion contradicts the statements in the introduction that "EPS production and bacterial utilization of newly produced EPS may decrease with increasing nutrient availability' (page 5, first lines). It is indeed generally assumed that extracellular release is a higher fraction of total primary production under nutrient-limiting conditions. On page 5 line 92-93 you write that you expected that increased nutrient availability would stimulate EPS production – I don't see why you would assume this, it is the opposite of what the literature suggests?

I feel the quantitative handling of the data is not always transparent or easy to follow. For the overall budgets in Figure 7, it is not clear to me how these were closed: you

have concentrations and d13C data on all these compartments, so you can calculate them individually – but they add up to 100% each time; you could add confidence to these numbers by verifying which % of the initial 13C-labeled biomass you can account for.

Figure 6: why are these first 'accounted for by loss of 13C in DIC & DOC" ? My first impression would be that you should simply look at the amount of 13C remaining in the sediment, without this 'correction' ? Please explain the rationale behind this in the text.

Towards the end of the discussion (line 704), the authors mention estimates of C retention at 30 days. This is odd, as the experiment ran over only 3.5 days and I would not consider extrapolations to 30 days very reliable (see also first comments).

Minor corrections

Abstract, line 15: what is meant with 'over-enrichment' ? I assume 'enrichment' suffices. Line 147: chlorophyll a (not alpha) Line 46-47: re-write this sentence, structure is odd.

―――――――――――――――――――――

---

## Author Comment (AC1) · 1 Mar 2018

We thank the two anonymous reviewers for their feedback. The suggested changes have helped to improve this manuscript.

The structure of the response is: 1) Reviewer comment number 1. Reviewer comment, 2. Author response 3. Changes to manuscript.

Highlighted text reflects portions of the text added as a result of comment.

Reviewer 1 main comments 1) 1. The authors applied a 13C label to exposed surface sediments at low tide and then waited 11 h to collect cores (lines 141-143, 144-145).

Approximately 2 h after collection, the cores were placed in the experimental tanks and exposed to the nutrient treatments (lines 150, 154-161). Carbon exchange, recycling, and loss between MPB and bacteria occurs quickly – over just a few hours. My concern is that at least 13 h passed between the 13C label addition and the application of the nutrient treatments. It seems that carbon exchange between MPB and bacteria that occurred prior to nutrient additions would confound the effects of the nutrient treatments.

2. Prior to incubation, all cores were maintained in conditions that were as close to identical as possible (both in situ and during transport) to maintain as much homogeneity as possible between the cores. As a result of this care, the carbon exchange between MPB and bacteria are expected to have been similar between cores prior to exposure to the nutrient additions as they were both handled in an identical manner and subsequently randomly allocated (LN 175) to treatment group prior to being placed in the incubation tanks.

As a result of this, carbon processing that occurred after the treatment application should be similar across cores within the treatments, regardless of the microbial processing that occurred in the 13 h prior to nutrient amendment. Any differences among treatments are therefore solely due to the application of different nutrient concentrations. The reviewer is correct that there was processing of newly fixed MPB-C that occurred prior to nutrient additions, but this manuscript focuses on the processing that occurred on the remaining MPB-C that was still labile and present (as evidenced by the 13C found within both bacteria and MPB) within the sediment during our incubations.

3. We have added a clarifying statement about the the effect of the 13 h prior to nutrient amendment. LN 168 now reads: "Processing of newly fixed MPB-C occurred in the 13 h prior to incubation with nutrient amendments, but was likely similar across cores as they were kept in identical conditions prior to incubation before random allocation to treatments. Although MPB-C was not freshly fixed at 13 h and likely more refractory as a result, the available C was still relatively labile and readily processed across all

treatments."

We have also clarified the focus of the study by more clearly stating the focus in the concluding paragraph of the introduction. LN 85 now reads: "In this 13C pulse-chase study we aimed to quantify the short-term effects of increased nutrient concentrations on the processing pathways for MPB-derived C within subtropical intertidal sediments"

2) 1. In addition, it is not clear from S1 that the nutrient treatments were effective.

2. As per text on ln 166-167, figure S1 was included to demonstrate that nutrient additions were incorporated into the biomass and were not sufficient to overload the capacity for nutrient utilization and leave a large concentration of DIN in the water column. We added nutrients in pulses that were processed and removed from the water column at both 0 d and 1.5 d. The sampling at 1.5 d was capped for incubation prior to nutrient addition and therefore was not expected to contain the nutrient addition that occurred at 1.5 d. Nutrient treatments were effective, as evidenced by the increased export of 13C from the sediment within the elevated treatment (LN 471), and the changes in distribution of labeled material as processing pathways were affected (Fig. 7).

3. No changes were made in response to this comment.

3) 1. Further, it seems from the methods that the cores in the same nutrient treatment tank all shared the same water column (lines 176-179). This would affect the independence of processes across cores and could confound the results.

2. The reviewer is correct that the water column was shared by all cores within each treatment tank. It was not logistically possible to separate paired cores into individual treatment tanks for each sampling and adding treatments to sealed cores would have exacerbated any 'bottle effect' given the relatively small volumes of water contained in each core.

Sharing a water column could have been a problem if 13C produced within one core found its way into another through transport of enriched DIC or DOC produced from

MPB-C. However, this was not a problem as the volume in each treatment tank was sufficient to dilute any label coming from these sources. We measured $\delta$13C of water column DIC and DOC; initial values for each light incubation (collected immediately after capping of cores) reflect the treatment tank water. No considerable enrichment in $\delta$13C values for initial DIC or DOC was observed for any of the sampling times across the 3.5 d incubations, confirming that there was no 13C transfer among cores within each treatment tank.

3. No changes were made in response to this comment.

4) 1. For the incubations, cores were sealed for 30 mins before initial samples were collected. Final samples were collected 3 h (light), 12 h (dark), or up to 16 h later (lines 186, 188-192). It would be good to clarify the exact duration of light and dark incubations and why the times differed between conditions.

2. The 3 h duration of the light incubation was intended to prevent dissolved oxygen from becoming supersaturated as a requirement for N2/Ar analysis that was being run simultaneously during this experiment as part of another study focused on N.

3. We have clarified that the incubation period was ~15 h (LN 188). LN 192 now reads: "Light incubations were of shorter duration to prevent supersaturation of dissolved oxygen which would have compromised additional analyses required for a complementary study."

5) 1. The authors should also clarify whether rates were calculated based on concentration changes between two time points. Generally, at least 3 time points are needed to calculate fluxes and capture nonlinear dynamics.

2. Several incubation studies have been performed by this working group across the last 15 years, and while they began with 3 to 5 point incubations (Ferguson et al. 2003, AME 33: 137-154, Veuger et al. 2007, L&O 52(5): 1930-1942) for flux measurements, the dynamics observed were largely linear. More recent studies have used two point

dark and light incubations due to the considerable reduction of cost and sampling effort (e.g., Oakes & Eyre 2014, Biogeosciences 11: 1927-1940) with minimal loss of information and accuracy.

3. We have clarified on LN 300: "Fluxes across the sediment-water interface were calculated from two measured concentrations, at the start and finish of each dark and light period (e.g., Oakes and Eyre 2014), as a function of incubation time, core water volume and sediment surface area."

6) 1. Moreover, long incubations, particularly under dark conditions, likely created 'bottle' effects that could affect metabolic rates. The authors should discuss potential artifacts of the sampling approach – particularly the potential impact of low DO on sediment respiration rates.

2. DO did not fall below 4.85 mg/L (DO% 58.5%) during the dark incubations, minimizing potential long term effects or 'bottle' effects. DO consistently increased rapidly when the lights were turned on, resulting in the need to shorten the light period in order to sample below supersaturation as required for N2/Ar analysis for another study occurring simultaneously with this one. The quick recovery of production indicates that any potential bottle effect during the longer incubation was negligible and did not result in considerable lag between respiration and production periods.

3. We have clarified that DO was not allowed either to drop too low or go above saturation. LN 310 now reads: "To prevent the potential development of resource limitations during incubation, O2 concentrations were not allowed to drop below 60% saturation in the dark and light incubations were shortened ($\sim$3 h) to ensure that production was not allowed to become supersaturated."

7) 1. Additional rationale for the time course of measuring fluxes 6 h after adding nutrients (and 19 h after adding 13C), and then again 1.5 d, 2.5 d, and 3.5 d would also be useful.

2. This project focused on initial processing of C, with multiple incubations over a relatively short period (3.5d after label addition), due to the observation in a comparable previous study at the same site that most 13C transformation occurs within this ∼4d (Oakes & Eyre 2014). Sampling was only feasible every ∼24 hours, with light/dark periods aligned with in situ conditions, and each light/dark incubation taking ∼15h. Acclimation time was allowed for any microclimates or disturbed anaerobic zonation to re-establish after disturbance with coring as this has the potential to influence P/R dynamics.

3. We have added a clarifying statement about the spacing of our sampling times. LN 204 now reads: "These sampling time periods were chosen to capture the active dynamics of 13C processing that were expected to occur over the first few days of the study, based on previous work by Oakes and Eyre (2014)."

We have added a clarifying statement about the rationale behind the 6 h acclimation period. LN 184 now reads: "Cores were allowed to acclimate in tanks for 6 h prior to the start of incubation to allow for the re-establishment of microclimates and anaerobic zonation that was potentially disturbed by coring."

8) 1. The authors need to provide more rationale for using 16:1n7 as a marker for diatoms. While this compound is an important component of diatom lipids, it is also produced by other groups of algae as well as by iron reducing bacteria and sulfate reducing bacteria. These communities are likely active in shallow coastal sediments and could be in close proximity to MPB if the sediments are anoxic. The absence of polyunsaturated C20 and C22 PLFAs seems to suggest that diatoms and other microalgae were at low abundances. I am not convinced that the current approach adequately isolates contributions from diatoms vs. non-diatoms.

2. We have used a previously published method (Oakes et al. 2016, L&O 61:2296-2308) for calculation of diatom associated 16:1n7. This calculation provides diatom associated 16:1n7 after subtraction of the estimated contribution of 16:1n7 from other

sources, calculated from 18:1n7 using a two source mixing model described on LN 280. 18:1n7 is a monosaturated PLFA that is associated with cyanobacteria, Gram negative bacteria, and sulfate reducing bacteria. By removing the amount of 16:1n7 estimated from 18:1n7, we removed the contribution of these bacterial groups from the total pool of 16:1n7. We did not investigate PLFAs associated with Fe-reducing bacteria as it was outside of the scope of this study. Furthermore, we are confident that pennate diatoms had substantial biomass at the site, and the contribution from other algal species was minimal based on visual analysis of the sediments (via microscopy) that confirmed strong dominance of pennate diatoms (LN 225). On this basis, we conclude that the remaining portion of 16:1n7 is predominately derived from diatoms.

3. We have clarified that 18:1n7 was used to account for the contribution of Gram negative, cyanobacteria, and SRB to 16:1n7 in the initial description of PLFA analysis. LN 227 now reads "The 16:1(n-7) PLFA, which represents 27.4% of total diatom PLFAs (Volkman et al. 1989), was consistently present across all samples and was used as a biomarker for diatoms, following correction for contributions from gram-negative bacteria, cyanobacteria, and sulfate reducing bacteria, determined using 18:1n7 as described in the calculations section below and in Oakes et al. (2016)."

9) 1. The carbon mass balance calculations should be clarified. For instance, the authors report the %C loss from the sediments, but it is unclear how this was calculated. Did this reflect the % change between the initial core collections and each sampling time point (0.5, 1.5, 2.5, and 3.5d)? Or was this calculated by subtracting out 13C losses via DIC and DOC (lines 490-491)? If the former, then that should automatically account for losses by respiration and exudation.

2. Neither method that the reviewer suggests reflects the accounting presented in this study.

Figure 6 shows 13C remaining in the sediment, not 13C loss. 13C remaining in the sediment was calculated as the sum of the exported DI13C and DO13C and sediment

derived 13C (bulk OC) divided by the sediment derived 13C and multiplied by 100 for each core at the end of each dark/light incubation. We then went on to use exponential decay functions to explore how the export of 13C from the sediment differed between nutrient amended treatments and ambient treatments.

We feel that we have adequately described how the turnover of 13C was calculated from our response to Reviewer 1 comment 10, briefly repeated here: "The data for 13C remaining in sediment OC were further examined by fitting an exponential decay function for each treatment across 3.5 d using the Exp2PMod1 function in OriginPro 2017 and 13C turnover estimates were then determined by solving for y = 0.05 % remaining 13C (a value close to 0) and x = 30 d for each treatment."

To further clarify, the %'s presented in the budget for Figure 7 reflect the portion of the 13C within each pool (MPB, Bacteria, uncharacterized) at each sampling time, with the interpolated fluxes of DIC and DOC included.

3. We have clarified our method of accounting for 13C on LN 261. "Percentages calculated from these pools are presented as portions of the sum of total 13C contained within the sediment and the interpolated fluxes of DIC and DOC that were estimated to have occurred from 0 d until each sampling time."

10) 1. Similarly it is unclear how the exponential decay functions were calculated.

2. Exponential decay functions were fitted using the function Exp2pmod1 in OriginPro 2017. 13C turnover estimates were provided by solving for y = 0 % remaining 13C and x = 30 d.

3. LN 350 now reads: "The data for 13C remaining in sediment OC were further examined by fitting an exponential decay function for each treatment across 3.5 d using the Exp2PMod1 function in OriginPro 2017 and 13C turnover estimates were then determined by solving for y = 0.05 % remaining 13C (a value close to 0) and x = 30 d for each treatment."

11) 1. The error bars in Figures 5 and 6 are large, particularly for the moderate and el-evated treatments; this variability makes the substantial differences across treatments (lines 475-478) somewhat unexpected.

2. Some of the error bars in Figs. 5 & 6 are large, as is expected for calculated fluxes across the multiple cores contained in each treatment. We have further investigated the 13C remaining in the sediment (Fig. 6) with a two-way ANOVA and found the elevated treatment to be significantly lower than both the ambient and minimal treatments. We have now included this analysis in our results.

To further investigate the differences between the loss rate constants presented in Fig. 6 resulting from the export of DIC and DOC shown in Fig. 5, we now present an analysis of the transformed data for 13C remaining in the sediment. This compared a model using a single slope for all of the data (i.e. including all treatments) with a model utilizing separate slopes for each treatment. We have added supplemental figure 2 that presents the slopes being compared as well as the underlying data being modeled.

3. LN 471 now describes the two-way ANOVA run for 13C remaining in sediments. LN 471 reads: "The total 13C remaining in sediment (Fig. 6) varied significantly among treatments (two-way ANOVA: $F_{3, 31}$= 5.7, p=0.008) and across sampling times ($F_{3, 31}$= 3.9, p=0.03). Throughout the study, there was generally less 13C remaining within the elevated treatment than in than either the ambient (p=0.008) or minimal treatments (p=0.02), and there was significantly less 13C remaining within the sediment at 3.5 d than at 0.5 d (p=0.02)."

LN 482 now describes how the comparison between slopes were analyzed: "Since the intercept is known, i.e., the initial value equals 100% at time 0, linear models where only the slopes were estimable, were fitted to further analyze the differences between slopes. Assuming an exponential decay, the percentage remaining 13C (Y) was log10 transformed and the value 2 was subsequently subtracted (Z=log10(Y) -2), which im-plies that the intercept of Z versus time equals 0. The model with different slopes

for each treatment fitted significantly better than the model with a single slope (F-test, $F_{3,28}=9.84$, P <0.001, Supplemental Fig. 2). The analysis was performed in R."

Supplemental Figure 2 has been added to LN 1030 along with a figure caption that reads: "Supplemental Figure 2: Slope comparison between treatments for log10 transformed 13C remaining in sediment. The model with different slopes for each treatment fitted significantly better than the model with a single slope (F-test, $F_{3,28}=9.84$, P <0.001)."

12) 1. The discussion should be more concise and focused. It would have been helpful if the authors considered processes driving variability across the treatments and compared their findings to other studies examining MPB-bacterial responses to surface water nutrient additions – particularly other stable isotope labeling experiments. Along these lines, I was somewhat surprised that nutrient additions would not stimulate MPB production and thereby promote C retention in the sediments.

2. We have examined the discussion and chosen to eliminate some included detail (previously LN 571-574 & LN 587-597) about patterns within the P/R data, as they were covered in too much detail in the text.

The reviewer's description of N stimulating MPB production describes only one side of the interaction that occurs within the intertidal biofilm. Opposite of MPB production, bacteria are working to actively hydrolyze both the EPS and biomass produced by MPB. Bacteria rely heavily on MPB-derived material for N under N-limitation. It may be possible that with increased N availability, bacteria are able to offset increased MPB production through increased hydrolysis and bacterial production. Figs. 3 and 4b support this, with increased incorporation of 13C into both bacteria and the uncharacterized pool (processed C) within the elevated treatments. It appears that the net balance between production and consumption of C within mudflat biofilms regulates whether there is net uptake or net loss of C within the sediment (Cook et al. 2007, Hardison et al. 2013, Spivak et al. 2016).

Labeling applications in coastal sediments with limited fauna and/or macroalgal influence that also include nutrient amendments are sparse in the literature. We have now included a comparison of retention of 13C within the sediment to Hardison et al. (2011). Given that there was very little fauna at our study site, and certainly no macrofauna within our incubated cores, we consider it inappropriate to directly compare to other studies such as Pascal et al. 2013 (6 year enrichment study for both C + N) or Spivak et al. 2016 (seasonal comparison using labeled detritus to work out MPB contribution) due to commentary within both studies about grazing pressure potentially masking bottom up effects of fertilization on MPB due to the faunal abundance found within the study site.

3. We have deleted a substantial amount of text from the discussion of P/R (previously LN 571-574 and LN 587-597).

LN 742 now includes a direct comparison to Hardison et al. 2011 for retention of 13C in sediment during a water column nutrient amended study.

Specific comments

Introduction: 13) 1. The authors acknowledge that other studies have examined the effects of nutrient loading on MPB (lines 73-77), but should discuss the subset of studies that used isotope tracer techniques in more depth, as these are directly relevant to the current manuscript.

2. The subset of studies that have used both biomarkers and labeled SI additions have not usually gone on to subsequently partition out the DIC and DOC portions of the carbon budget and therefore do not fully discuss these processing pathways. To fully address both portions of the budget that this study examines (sediment and DIC/DOC), we have broadened our focus and included a combination of references for both tracer studies and non-tracer studies that investigated nutrients and MPB on LN 80-84: "Increased autochthonous production driven by nutrient enrichment can lead to increased heterotrophy, as newly produced organic matter is mineralized (Fry et

al. 2015), resulting in increased DIC production. Increased remineralization of newly produced MPB-C will result in greater loss of DIC from intertidal sediment via bacterial respiration (Hardison et al. 2011)." The reason for this diverse focus is that most tracer studies utilizing biomarkers focus solely on partitioning of the budget directly involved with those biomarkers and do not present accounting of the DIC and DOC pathways as a portion of the budget within the framework of the study. Additionally, we have added Pascal et al. (2013) as another relevant stable isotope study in addition to Cook et al. (2007) which was already included (a tracer study examining the effects of nutrient enrichment) and discussed on LN 78: "Both EPS production and bacterial utilization of newly produced EPS may decrease with increasing nutrient availability". 3. We have added the reference to Pascal et al. (2013) to LN 76.

14) 1. Line 113 and elsewhere: instead of reporting carbon concentration per surface area, please report % organic carbon.

2. We have now provided sed %OC in the study site description in addition to our previous units. Elsewhere in the manuscript we have retained units as per surface area as it is convention for work of this nature (e.g. Oakes et al. 2014, Hardison et al. 2011, and Middelburg et al. 2000) and allows for comparison across studies as well as conversion into %OC if anyone should wish to do so.

3. %OC is now included on LN 113 in addition to the previously provided units for concentration per surface area.

15) 1. Lines 155-160. What were the target concentrations of each of the treatments and was the site water filtered before the treatments were applied?

2. LN 155 reads: "The treatment tanks were set up at ambient concentration (site water, DIN of $2.5 \pm 0.04$ $\mu$mol N L-1, measured on incoming tide), and with N ($NH_4+$) and P ($H_3PO_4$) amendment to unfiltered site water at $2\times$ (minimal treatment), $5\times$ (moderate treatment) and $10\times$ (elevated treatment) average water column concentrations near the study site (4 $\mu$mol L-1 $NH_4+$ and 5 $\mu$mol L-1 $PO_43-$; Eyre (1997; 2000))."

The target concentrations for pulsed nutrient amendments were for 8, 25, and 40 $\mu$mol L-1 NH4+ and 10, 25, and 50 $\mu$mol L-1 PO43- respectively for minimal, moderate and elevated treatment pulses. Site water was unfiltered.

3. We have now specified that the water was not filtered (LN 155). No changes were made to further specify the target concentrations of each treatment, as these can clearly be derived from the text on LN 155 (2x, 5x, and 10x average water column concentrations of 4umol/L NH4 and 5umol/L PO4).

Calculations: 16) 1. The biomass calculations use several conversion factors to scale from lipid concentrations to bacterial and diatom biomass. It is unclear how well constrained the conversion factors are and the amount of uncertainty they introduce into the calculations. For instance, does the concentration of lipid per unit biomass change with algal growth or nutrient condition? It is unclear why the lipid concentrations need to be scaled to biomass – why not compare the ratio of bacterial to microalgal lipids?

2. This comment addresses concerns about methodology that are not unique to this study, but which are common to methods routinely applied in studies using biomarkers. Conversion factors are frequently applied to FAMEs (fatty acid methyl esters) to estimate and compare biomasses of different microbial groups (bacteria, MPB).

When examining uptake of 13C, we need to account differences in biomass to adequately account for the total 13C present in different pools. It is possible to have a large enrichment of 13C contained in a small pool of biomass that may contain less 13C than a minimally enriched pool of biomass that is considerably larger. Without accounting for the relative biomass that is contained within each pool, it would appear the more enriched biomarker is more involved in the processing of 13C, but upon full accounting of both biomass and enrichment, it becomes abundantly clear that the larger, less enriched pool has a more significant impact.

As such, thorough investigation of "the concentration of lipid per unit biomass change with algal growth or nutrient condition" falls outside of the scope of this study.

3. We have referred to previous work that utilize these conversion estimates on LN 227: "The 16:1(n-7) PLFA, which represents 27.4% of total diatom PLFAs (Volkman et al. 1989), was consistently present across all samples and was used as a biomarker for diatoms, following correction for contributions from gram-negative bacteria, cyanobacteria, and sulfate reducing bacteria, determined using 18:1n7 as described in the calculations section below and in Oakes et al. (2016)."

17) 1. Equations 3-4 are descriptions more so than equations.

2. The reviewer is technically correct, but these serve to efficiently communicate how the fluxes for both respiration and net primary productivity were calculated in order to further calculate gross primary productivity.

3. We consider the current presentation to be both clear and concise; no changes have been made.

18) 1. Line 330: ANOVA is not the most appropriate test because the cores within a treatment were not independent of one another as they shared a water column.

2. As stated in the previous response to Reviewer 1 comment 3: "Sharing a water column could have been a problem if 13C produced within one core found its way into another through transport of enriched DIC or DOC produced from MPB-C. However, this was not a problem as the volume in each treatment tank was sufficient to dilute any label coming from these sources. We measured $\delta$13C of water column DIC and DOC; initial values for each light incubation (collected immediately after capping of cores) reflect the treatment tank water. No considerable enrichment in $\delta$13C values for initial DIC or DOC was observed for any of the sampling times across the 3.5 d incubations, confirming that there was no 13C transfer among cores within each treatment tank."

3. The reviewer does not provide an alternative analysis for us to investigate, so we have considered their case for a lack of independence and provided refuting evidence. No changes were made as a result of this comment.

19) 1. Line 341: What is the ecological rationale for grouping time points 0.5 and 1.5 vs. 2.5 and 3.5?

2. As stated in the text on LN 341: "To increase replication for statistical analysis, and therefore increase the power to detect a significant difference, we therefore grouped data across times into two levels: before 1.5 d (including 0.5 d and 1.5 d) and after 1.5 d (including 2.5 d and 3.5 d).", the rationale is largely statistically driven. The division provides an equal group weighting between the examined groups. There is some geochemical rationale in doing this as the additions of tracers are often observed to change the most in the earliest part of the experiment, and taper off as the carbon becomes more recalcitrant and dispersed within the system, but primarily, this split was designed to allow for sufficient replication within the groups and, therefore, robust statistical analysis.

3. No changes were made as a result of this comment.

20) 1. Lines 415-426. How did downward transport occur in these cores? MPB are generally restricted to the top 2 cm and it does not seem that there was pore water flow during the lab incubations. Was there mixing by animal communities? Is it possible that contamination occurred during core collection in the field?

2. Saburova et al. 2003 examined diatom migration in detail and found 4 cm migration in sediment with clay sublayers, and up to 8 cm migration in sediments with a coarse sand sublayer, so the reviewer's statement that MPB migration is limited to 2 cm is incorrect. As our site sediments were quite sandy, and the stir bars suspended above the sediments were sufficient to stimulate moderate porewater flow in the upper layer, we consider it plausible that downward transport occurred via MPB transport across depth during the incubations. Saburova et al. 2003 is referenced in the downward transport section of the discussion (LN 717) supporting MPB migration as an effective pathway for downward transport in this study.

Contamination is possible during collection, but would have been minimal and fairly

uniform across the cores since the core liners were placed into the sediment in an identical manner. Additionally, our initial cores displayed minimal downward presence of 13C, which also supports that contamination was minimal during collection as they were collected in exactly the same manner as sample cores.

3. No changes were made as a result of this comment.

21) 1. Line 457: Was the uncharacterized fraction defined as PLFAs that were not i,a-15 or 16:1n7? If so, additional rationale is needed to justify this approach.

2. The 13C content of the uncharacterized fraction was calculated by subtracting the 13C in contained microbial biomass (diatom and bacteria calculated from PLFAs) from the 13C contained in the sediment organic carbon. This is now described in the methods section.

3. LN 295 now reads: "Microbial biomass is the sum of calculated diatom and bacterial biomass. Uncharacterized 13C was calculated as: 3. 13Cuncharacterized = 13Csediment organic – 13Cmicrobial biomass where 13Csediment organic represents total 13C in sediment organic carbon and 13Cmicrobial biomass represents the 13C contained in microbial biomass within the same core."

22) 1. Graphs: It would be helpful if the error bars were positive and negative

2. We attempted to add both positive and negative errors bars, but with the complexity of the experimental design that is presented, positive and negative error bars achieve little other than cluttering the bar graphs being presented. This is especially true for figures 1A and 2 where both positive and negative fluxes are presented and with several pools being displayed in the same figure.

3. No changes were made as a result of this comment.

[Figure]

**Fig. 1.**

---

## Author Comment (AC2) · 1 Mar 2018

We thank the two anonymous reviewers for their feedback. The suggested changes have helped to improve this manuscript.

The structure of the response is: 1) Reviewer comment number 1. Reviewer comment, 2. Author response 3. Changes to manuscript.

Highlighted text reflects portions of the text added as a result of comment.

Reviewer 2 main comments

1) 1. The manuscript by Riekenberg et al. describes data from 13C-incubation experiments whereby microphyobenthos was labeled with 13C in situ, and then incubated under controlled conditions over a period of 3.5 days; either with background nutrient levels or with higher than ambient N &/or P concentrations. While generally a well performed study, I am surprised by the short duration of the experiments (3.5 days). When fitting exponential decay functions on the resulting data (Figure 6), I feel this is somewhat thin ice – the data should be spread more in time for a convincing exponential fit.

2. The reviewer is correct, the short time frame and relatively few data points means that the rate of loss may not be entirely representative of longer time loss at the site. However, the aim of this part of the study was to determine the relative differences in loss rates between the nutrient amended treatments and the ambient treatment. The difference between treatments is clear (now supported with a two-way ANOVA on LN 471) and similar whether an exponential or linear fit is required. The use of an exponential relationship was supported given that previous studies using a similar method of 13C application have found robust exponential relationships when describing the loss of 13C from sediment over a longer period, including data over the first few days following labeling (Oakes et al. 2014 Fig. 4; Veuger et al. 2012). Oakes et al. 2014 is a labeling study that occurred in the intertidal range within this same site that found exponential loss to be adequate to model 13C loss from this site.

In response to the 'thin ice' comment, we have more thoroughly investigated the differences in sediment retention of 13C between treatments using a two-way ANOVA. This analysis indicated significant differences between the means for the elevated and both the ambient and minimal treatments. We subsequently explored the relationships between treatments and found the difference observed between the elevated and ambient treatments to be robust as demonstrated by the analysis now provided (LN 471) that examines the differences between the slopes of loss found for each treatment (Supplemental Figure 2). It is also interesting to note the apparent dose-effect relationship within the data set, as the moderate treatment appears to fall in between that of the

minimal and elevated treatment.

Regarding the short duration of this study, we have addressed this in a previous comment from Reviewer 1 comment 7 briefly repeated here: "This project focused on initial processing of C, with multiple incubations over a relatively short period (3.5d after label addition), due to the observation in a comparable previous study at the same site that most 13C transformation occurs within this ∼4d (Oakes & Eyre 2014)."

3. As a result of this comment, we have now performed a two-way ANOVA on the % of 13C retained in the sediment provided on LN 471 and we have also included an analysis comparing the differences between slopes for our loss rates. LN 471 now reads: "The total 13C remaining in sediment (Fig. 6) varied significantly among treatments (two-way ANOVA: F3, 31= 5.7, p=0.008) and across sampling times (F3, 31= 3.9, p=0.03). Throughout the study, there was generally less 13C remaining within the elevated treatment than in than either the ambient (p=0.008) or minimal treatments (p=0.02), and there was significantly less 13C remaining within the sediment at 3.5 d than at 0.5 d (p=0.02)."

Our response to Reviewer 2's second comment also further addresses the relevance of the turnover times calculated from this study and how they compare to previous labeling experiments performed in similar environments.

We have also added a clarifying statement about the duration of the study in response to Reviewer 1 comment 7 repeated here: LN 204 now reads: "These sampling time periods were chosen to capture the active dynamics of 13C processing that were expected to occur over the first few days of the study, based on previous work by Oakes and Eyre (2014)."

2) 1. Abstract, line 30 and in Discussion: clearly define in the manuscript how you define and calculate the turnover time, to avoid any ambiguity. I find these turnover times surprisingly high (i.e., long), and in line with other comments, wonder whether the short incubation period did not lead to a bias in this estimate – with 3 time points

very early on it seems not ideal to fit an exponential fit to these data.

2. We disagree with the reviewer that turnover times are high. Our estimates of retention at 30 d (18-58%, LN 757) fall around the range found for other studies in unvegetated sediments (30-50%) as stated on LN 740:

"Although the focus has primarily been on vegetated environments (Duarte et al. 2005), which store the most carbon, unvegetated sediments also have capacity for longer-term retention (e.g. $\sim$50% after 21 d Hardison et al. 2011, 30% after 30 d Oakes and Eyre 2014; 31% after 30 d Oakes et al. 2012)."

The relatively short incubation times were ideal to explore the short-term fate of MPB-C which was the primary focus of this study. We then tried to use that data to see what implications these dynamics may have for longer-term retention. This is clearly not ideal, as it requires extrapolation from a small data set, but the relative differences between treatments are robust. Future studies examining nutrient effects on 13C retention should examine this relationship across a longer time period if possible.

3. We have now clearly defined how we have calculated turnover time with the inclusion of our clarifying statement on LN 350: "The data for 13C remaining in sediment OC were further examined by fitting an exponential decay function for each treatment across 3.5 d using the Exp2pMod1 function in OriginPro 2017 and 13C turnover estimates were then determined by solving for y = 0.05% remaining 13C (a value close to 0) and x = 30 d for each treatment." We have included a statement supporting our utilizing exponential functions on LN 527: "The focus of this study was short-term fate, but our findings also show potential implications for longer- term retention. Our calculated retention times may be under or over-estimated due to their reliance on short-term data. However, the relative differences between treatments (decreased retention with increased nutrient amendment) are clear. The rationale for utilizing exponential functions in this study follows previous findings in Oakes et al. (2014) that 13C export from subtidal sediments at this site were well-described by an exponential decay function

across a longer time period (31 d). Additionally, the 30 d estimates provided within this study (18-58%) fall across a range similar to that of other previous labeling experiments (30-50%; Hardison et al. 2011; Oakes and Eyre 2014; Oakes et al. 2012), leading the authors to conclude that the use of exponential functions to describe this relationship was valid in this study."

3) 1. Also, it is not unambiguously clear what your t=0 is (after the 6 hour 'acclimation period' ? See next comment). On page 9, line 184, the authors mention that the cores were allowed to 'acclimate for 6 hrs prior to the start of the incubation'. I'm not sure what this means, it's not as if no microbial activity would take place during this period, hence for me it would seem to be an integral part of the incubation period. Why not simply define t=0 as the moment the cores were no longer exposed to 13C-DIC labeling ? Are these 6 hours part of the incubation times mentioned throughout the ms ? If not, this may bias the estimates of turnover times.

2. The purpose of the acclimation period was to allow for re-establishment of any disturbed sediment redox zonation that occurred during coring (see changes due to reviewer 1 comment 7). Therefore, we omitted the 6 h period prior to measuring in an effort to obtain more robust measurements for P/R and water column fluxes for DOC and DIC. As this study was focused primarily on the development of differences between the treatments kept under the same conditions, we do not agree with the reviewer that this 6 h period is integral to the results presented.

The described shift used with the exponential functions from the current 0 to 19 h before (when the 13C was at its maximum) describes the mathematical parameter of an x transform shift by 0.8 d. Calculation of the resulting change results in materially the same estimates for turnover time (shifted longer by 0.8 d). There is no apparent bias within the dataset resulting from this as the functions used to model loss rates were not forced to 100% at the starting point (x=0).

3. This comment further supports the previous changes that resulted from Reviewer

1, comment 7, copied here: We have added a clarifying statement about the rationale behind the 6 h acclimation period. LN 184 now reads: "Cores were allowed to acclimate in tanks for 6 h prior to the start of incubation to allow for the re-establishment of microclimates and anaerobic zonation that was potentially disturbed by coring."

4) 1. In the abstract (line 26-27), the authors mention that treatments with higher nutrient levels showed higher loss of 13C label, "supporting increased production of extracellular enzymes and storage products". I have two reservations here: First, this pattern would equally be consistent with a scenario in which the heterotrophic bacterial community was N and/or P-limited ? Eg Keuskamp et al. Sci Total Environ. 2015 doi: 10.1016/j.scitotenv.2014.11.092. I would suggest to add this as a possible mechanism in the introduction on page 4 (section starting at line 67).

2. We agree with the reviewer that relaxation of nutrient limitation may have increased the extracellular products being released. We previously postulated that relaxation of nutrient limitation had potential to affect microbial biomass on LN 67.

3. We have added this possibility to our previous statement in LN 67: "A major source of environmental change in coastal systems is nutrient over-enrichment (Cloern et al. 2001), which may affect the assimilation and flux pathways of MPB-derived carbon through 1) increased microbial biomass or an increase in production of extracellular enzymes resulting from relaxation of nutrient limitation, 2) increased algal production that drives elevated heterotrophic processes as bacteria utilize newly produced C, and 3) increased loss of C as DIC via respiratory pathways as heterotrophic processes."

5) 1. Secondly, this conclusion contradicts the statements in the introduction that "EPS production and bacterial utilization of newly produced EPS may decrease with increasing nutrient availability' (page 5, first lines). It is indeed generally assumed that extracellular release is a higher fraction of total primary production under nutrient-limiting conditions. On page 5 line 92-93 you write that you expected that increased nutrient availability would stimulate EPS production – I don't see why you would assume this, it

is the opposite of what the literature suggests?

2. The reviewer is correct that the fraction of primary productivity that EPS represents is higher in nutrient limited settings as MPB produce EPS as a way to manage excess C. We agree that the nutrient addition should stimulate overall MPB-C production and not just that of EPS and have changed the wording to reflect that.

3. LN 92 now reads: "We expected increased nutrient availability to stimulate production of MPB-C after initial labeling, resulting in decreased turnover times for MPB-C as well as a shift towards dominance of heterotrophic processes as bacteria utilize this additional labile C"

6) 1. I feel the quantitative handling of the data is not always transparent or easy to follow. For the overall budgets in Figure 7, it is not clear to me how these were closed: you have concentrations and d13C data on all these compartments, so you can calculate them individually – but they add up to 100% each time; you could add confidence to these numbers by verifying which % of the initial 13C-labeled biomass you can account for.

2. We closed the budgets in Figure 7 by accounting for the total 13C contained in the bulk sediment organic C and including the calculated fluxes for DIC and DOC that were interpolated from measurements across the 3.5 d for both DIC and DOC.

3. To make it clear how the budgets were constructed and closed, we have included a brief statement in the methods section about how the budget was calculated. LN 326 now reads: "Because all 13C was contained within the cores, values for 13C budgets add to 100%. Starting values were estimated by looking at how much 13C remained in the sediment and how much was lost to the water column (initial 13C= 13C remaining + 13C lost)."

7) 1. Figure 6: why are these first 'accounted for by loss of 13C in DIC & DOC" ? My first impression would be that you should simply look at the amount of 13C remaining

in the sediment, without this 'correction'? Please explain the rationale behind this in the text.

2. Since these core incubations are a contained system, the estimate of original 13C present adds to 100%, with the measured fluxes of DIC and DOC needing to be incorporated in order to correctly portion the amount of 13C still contained in the sediment versus the amount exported to the water column across the incubation period.

3. We have further clarified how the 13C budgets were constructed as a result of Reviewer 2 comment 6, briefly repeated here: To make it clear how the budgets were constructed and closed, we are including a brief statement in the methods section about how the budget was calculated. LN 326 now reads: "Because all 13C was contained within the cores, values for 13C budgets add to 100%. Starting values were estimated by looking at how much 13C remained in the sediment and how much was lost to the water column (initial 13C= 13C remaining + 13C lost)."

8) 1. Towards the end of the discussion (line 757), the authors mention estimates of C retention at 30 days. This is odd, as the experiment ran over only 3.5 days and I would not consider extrapolations to 30 days very reliable (see also first comments).

2. We address this with quoted text in reviewer 2's comment 2, but will expand further here. On LN 740, we detail findings of other studies that determined 13C retention across 30 days. Further in that paragraph (LN 757) we provide our estimates of retention at 30 d taken from our exponential functions. Our ambient and minimal estimates fall close to the range of these previous studies. This finding allows us to feel reasonably confident that we are not too far out of the ballpark working with our exponential fits in the data set. Being able to gauge our results against the findings of other studies that extended over this period (e.g. Oakes & Eyre 2014) was the goal behind extrapolating our data to 30 d. We realize that extrapolation is not terrifically reliable when based on a data set formed over a short time period, but extrapolation in this case allows us to compare our short-term loss rates to other longer-term studies in an effort

to gauge whether the rates we have found agree and are in a reasonable range.

3. A clarifying statement about the 30 d comparisons is now provided, pointing out that substantial extrapolation is required in order for us to compare to other studies looking at longer term retention of MPB-C. LN 752 now reads: "The primary focus of this study was short-term fate of MPB-C, but the significant decrease in retention observed with nutrient amendments imply that short-term processes may have implications for longer term retention. It is interesting to consider how these short-term changes may affect the longer-term retention (30 d) reported by previous studies (e.g., Oakes & Eyre 2014), with the caveat that the substantial extrapolation required could introduce considerable error to estimates of retention."

Minor corrections 9) 1. Abstract, line 15: what is meant with 'over-enrichment' ? I assume 'enrichment' suffices.

2. Enrichment does suffice, corrected as suggested.

10) 1. Line 148: chlorophyll a (not alpha)

2. Fixed throughout manuscript.

11) 1. Line 46-47: re-write this sentence, structure is odd.

3. LN 46 now reads: "Application of rare isotope tracers can render fractionation effects and variability that affect natural abundance stable isotope techniques negligible and has been useful for elucidating pathways for the processing and loss of MPB-derived C within estuarine sediments."
* * *
[Figure]

**Fig. 1.**